# A Closer Look at TabPFN v2: Understanding Its Strengths and Extending Its Capabilities

**Han-Jia Ye & Si-Yang Liu**
School of Artificial Intelligence, Nanjing University
National Key Laboratory for Novel Software Technology, Nanjing University
{yehj, liusy}@lamda.nju.edu.cn

**Wei-Lun Chao**
The Ohio State University
chao.209@osu.edu

## Abstract

Tabular datasets are inherently heterogeneous, presenting significant challenges for developing pre-trained foundation models. The recently introduced transformer-based Tabular Prior-data Fitted Network v2 (TabPFN v2) achieves unprecedented *in-context learning* performance across diverse downstream datasets, marking a pivotal advancement in tabular foundation models. In this paper, we take a closer look at TabPFN v2 to examine how it effectively handles heterogeneity and achieves high predictive accuracy, and to explore how its limitations in high-dimensional, many-category, and large-scale tasks can be mitigated. We find that TabPFN v2 can infer attribute relationships even when provided with randomized attribute token inputs, eliminating the need to explicitly learn dataset-specific attribute embeddings to address heterogeneity. We further show that TabPFN v2 can be transformed into a feature extractor, revealing its ability to construct a highly separable feature space for accurate predictions. Lastly, we demonstrate that TabPFN v2's limitations can be addressed through a test-time divide-and-conquer strategy, enabling scalable inference without requiring re-training. By uncovering the mechanisms behind TabPFN v2's success and introducing strategies to extend its applicability, this study offers key insights into the design of future tabular foundation models.

## 1 Introduction

Tabular data is ubiquitous across a wide range of applications, including healthcare [32], finance [43], and scientific research [33, 32]. In this format, each instance (*e.g.*, a patient's record) is represented as a vector of attributes, and the goal of a machine learning model is to map these vectors to their corresponding labels [7]. Traditionally, tree-based models [56, 12] have dominated this domain, but recent advances in deep tabular models are increasingly closing the performance gap [22, 29, 76].

However, unlike vision and language domains, where pre-trained foundation models have driven significant progress [40, 81], tabular data is still desperately awaiting a similar breakthrough [63, 27, 54, 82, 77, 66]. *A primary challenge arises from the inherent heterogeneity of tabular datasets, which often vary in dimensionality and attribute meanings, making the development of effective and versatile foundation models difficult.* Additionally, there is an urgent need for such models, as many tabular datasets are small-scale—such as medical data with limited patient numbers. Training individual models from scratch for these datasets is highly sensitive to hyperparameter choices and often fails to generalize due to limited data [18, 24, 25].

39th Conference on Neural Information Processing Systems (NeurIPS 2025).

Recently, the Tabular Prior-Fitted Network v2 (TabPFN v2) [28] has emerged as a significant step forward. Built on transformer architectures [67] and pre-trained on gigantic synthetic datasets [27, 28], TabPFN v2 can be directly applied to diverse downstream tasks without additional tuning. Specifically, TabPFN v2 takes both a labeled training set and an unlabeled test instance as input, predicting the test label in an "in-context learning" manner. When evaluated across both classification and regression tasks, TabPFN v2 consistently outperforms prior tabular methods, achieving state-of-the-art accuracy.

Motivated by the remarkable performance of TabPFN v2, we aim to take a step further to understand the mechanisms behind its success[1]—specifically, how it effectively handles dataset heterogeneity and achieves high predictive accuracy. In addition, we investigate how to overcome its current limitations—namely, its suggested data regime of no more than 10,000 samples, 500 dimensions, and 10 classes [28]—ideally without requiring model re-training. We outline major insights as follows.

1. **TabPFN v2 internalizes attribute token learning to handle data heterogeneity.** Given an instance with $d$ attributes, TabPFN v2 transforms it into a set of fixed-dimensional tokens and uses a transformer architecture to handle variability in $d$, following [64, 22, 74]. In sharp contrast to prior methods that rely on known attribute semantics (*e.g.*, word vectors) or learn dataset-specific attribute tokens, TabPFN v2 instead employs randomized attribute tokens—resampled at each inference. This design "syntactically" allows TabPFN v2 to be directly applied to new downstream datasets with varying dimensionalities and attribute meanings without additional tuning, but raises a fundamental question: how does it still make accurate predictions? Our analysis shows that, regardless of the randomness, TabPFN v2 can consistently infer attribute relationships through in-context learning, essentially integrating attribute token learning into the inference itself. In short, TabPFN v2 unifies representation learning and prediction within a single forward pass.

2. **TabPFN v2 can be repurposed as a feature extractor for downstream tasks.** The exceptional predictive performance of TabPFN v2 suggests that it produces instance-level feature representations that are highly discriminative. However, verifying this is non-trivial, as TabPFN v2's in-context learning mechanism assigns distinct roles to labeled training and unlabeled test instances, resulting in embeddings that are not directly comparable. To overcome this, we propose a leave-one-fold-out strategy that enables the extraction of instance features more closely aligned across training and test data. Our findings reveal that TabPFN v2 effectively maps tabular instances into a nearly linearly separable embedding space. Remarkably, training a linear model on these features yields accuracy comparable to that of TabPFN v2's in-context learner, highlighting its potential as a powerful feature encoder. This not only offers insights into TabPFN v2's inner workings but also opens the door to broader applications (*e.g.*, visualization and error analysis).

3. **Test-time divide-and-conquer effectively mitigates TabPFN v2's limitations.** As noted in [28], TabPFN v2 faces challenges when applied to high-dimensional, many-category, or large-scale datasets. Rather than resorting to model re-training, we show that these limitations can be effectively addressed through carefully designed *post-hoc* divide-and-conquer strategies, reminiscent of test-time scaling techniques developed for large language models [70, 51]. Empirical results show significant accuracy gains across these challenging data regimes, highlighting the potential of advanced post-hoc methods to further extend the capabilities of tabular foundation models.

**Remark.** This paper presents a timely and in-depth investigation into TabPFN v2, offering valuable insights for advancing tabular foundation models. While we do not propose a new architecture or training scheme, our contribution lies in the novel analysis and principled extension of TabPFN v2. This reflects a growing trend in foundation model research, where understanding, evaluating, and adapting powerful models is increasingly seen as being as impactful as designing new ones.

## 2 Related Work

**Tabular foundation models.** Pre-trained models have revolutionized the vision and language domains [40, 81], but their adoption in tabular data remains limited due to the substantial heterogeneity across datasets. Variations in attribute spaces, dimensionalities, and label distributions present significant challenges for joint training and transferability. One solution is to leverage the semantic meanings of attributes, as demonstrated by methods that convert tabular instances into textual descriptions and

---

[1]We note that TabPFN v2 [28], like many recent large language models (LLMs) and foundation models, does not release its training data or training recipe. Accordingly, our focus is on understanding the properties of the released pre-trained model and exploring ways to extend its applicability.

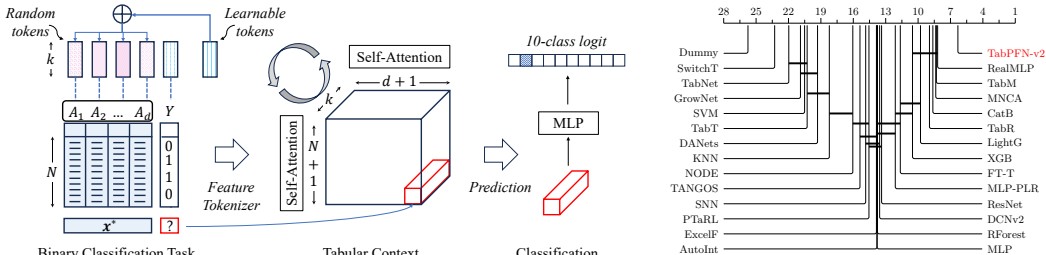

Figure 1: **Left**: Illustration of TabPFN v2's mechanism for binary classification [28]. $\{A_1, \ldots, A_d\}$ denote $d$ attributes of the task. Training examples and a test instance are combined into a tabular context and transformed into a $(N + 1) \times (d + 1) \times k$ tensor using a combination of learnable and randomized tokens. Two types of self-attention are applied alternately across rows (inter-sample) and columns (inter-feature). The output token corresponding to the (dummy) label of the test instance is processed through an MLP to generate a 10-class logit. **Right**: Wilcoxon-Holm test at a significance level of 0.05 over 273 small- to medium-scale datasets. We omit the 27 datasets used to select TabPFN v2's checkpoint from the 300 datasets in [75].

apply large language models for prediction [26, 79, 69, 71]. Alternatively, some approaches aim to improve transferability by pre-computing attribute tokens based on semantic embeddings [74, 39]. In practical domains such as healthcare or scientific measurement, the semantic meanings of attributes are often inaccessible due to privacy constraints, annotation costs, or a lack of describability. To address this, [77] proposed representing each instance by its similarity profile to a fixed number of nearest neighbor examples, thereby mapping it into a consistent latent space with shared dimensional semantics. The TabPFN family [27, 28] leverages the in-context learning capabilities of transformers to directly predict labels by contextualizing test instances among training examples. This strategy inspired subsequent pre-trained tabular models such as [48, 15, 57]. While TabPFN v1 pads attribute vectors to a fixed dimension, TabPFN v2 introduces a specialized attribute tokenizer to handle heterogeneous input spaces. Meta-learning has also been explored to generate model weights tailored for downstream tabular tasks with limited data [34, 6, 50]. Other pre-trained models rely on lightweight fine-tuning to adapt to variations in attribute and label spaces [44, 80, 62, 82].

## 3 Background

**Learning with a single tabular dataset**. A tabular dataset $\mathcal{D} = \{(\boldsymbol{x}_i, y_i)\}_{i=1}^N$ contains $N$ training examples, corresponding to the rows in a table. Each instance $\boldsymbol{x}_i$ is characterized by $d$ features or attributes (*i.e.*, columns in the table), where $d$ typically varies across datasets. Its label $y_i$ belongs to $[C] = \{1, \ldots, C\}$ for a classification task or is a numerical value for a regression task. We assume that all attributes of an instance are numerical (continuous). Categorical (discrete) attributes, if present, are transformed using ordinal or one-hot encoding beforehand. The goal of tabular machine learning is to learn a mapping $f$ from instances to their labels. Specifically, given an unseen instance $\boldsymbol{x}^* \in \mathbb{R}^d$ sampled from the same distribution as $\mathcal{D}$, the learned mapping $f$ predicts its label as $\hat{y}^* = f(\boldsymbol{x}^* \mid \mathcal{D})$. A smaller discrepancy between $\hat{y}^*$ and the true label $y^*$ indicates stronger generalizability of $f$.

**TabPFN**. The original TabPFN implements $f$ for classification using a transformer-like architecture [27]. Both training and test instances are first *zero-padded* to a fixed dimension $k'$ (*e.g.*, 100). Then, $\boldsymbol{x}_i$ and $y_i$ are linearly projected to $\tilde{\boldsymbol{x}}_i \in \mathbb{R}^k$ and $\tilde{\boldsymbol{y}}_i \in \mathbb{R}^k$, respectively. TabPFN processes both a labeled training set and an unlabeled test instance jointly, predicting the test label in an *in-context learning* manner. The task context is defined as $\mathcal{C} = \{(\tilde{\boldsymbol{x}}_1 + \tilde{\boldsymbol{y}}_1), \ldots, (\tilde{\boldsymbol{x}}_N + \tilde{\boldsymbol{y}}_N), (\tilde{\boldsymbol{x}}^*)\} \in \mathbb{R}^{(N+1) \times k}$, consisting of $N + 1$ tokens, each of dimension $k$. These tokens are processed by multiple transformer layers, which accommodate variable-length inputs (*i.e.*, variable $N$). The output token corresponding to the test instance is passed through a multi-layer perceptron (MLP) to produce a 10-class logit.

**TabPFN v2**. The recently proposed variant [28] introduces several key modifications. First, each of the $d$ attributes in $\mathcal{D}$ is embedded into a $k$-dimensional space, with random perturbations added to differentiate attributes. Together with the label embedding $\tilde{\boldsymbol{y}}_i \in \mathbb{R}^k$, each training instance $\boldsymbol{x}_i$ is represented by $(d + 1)$ tokens with dimension $k$. For a test instance $\boldsymbol{x}^*$, where the label is unknown, a dummy label (*e.g.*, the average label of the training set) is used to generate the label embedding $\tilde{\boldsymbol{y}}^*$.

The full input to TabPFN v2—comprising the training set and the test instance—is thus represented as a tensor of shape $(N + 1) \times (d + 1) \times k$. Two types of self-attention are applied in alternation: one over samples (among the $N+1$ instances) and the other over attributes (among the $d+1$ dimensions),

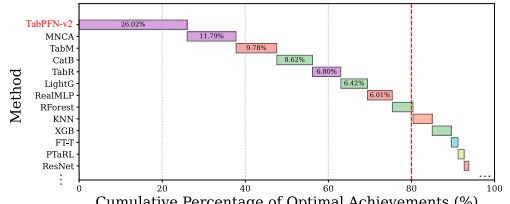

Figure 2: Probability of Achieving the Maximum Accuracy or Minimum RMSE across 273 datasets. Values inside rectangles show the percentage of datasets on which a method achieves the best result.

| $\downarrow$ | TabPFN v2 | CatB | MNCA | R-MLP | LR |
|---|---|---|---|---|---|
| High-Dim | 3.36 | 2.82 | 4.41 | 2.14 | 2.27 |
| Large-Scale | 3.97 | 1.89 | 2.27 | 1.94 | 4.47 |
| >10 classes | 3.33 | 2.75 | 3.17 | 1.42 | 4.33 |

Table 1: Average rank (lower is better) of TabPFN v2 and representative baselines on 18 high-dimensional, 18 large-scale, and 12 datasets with more than 10 classes. Full results with our extensions are in Figure 5.

enabling in-context learning along both axes. Finally, the output token corresponding to the test instance's dummy label $\tilde{y}^*$ is extracted and mapped to a 10-class logit for classification or a single-value logit for regression. An overview of this process is illustrated in Figure 1 (left).

The weights in TabPFN v2 are pre-trained on diverse synthetic datasets generated using structural causal models (SCMs), with the checkpoint selected based on real-world datasets. For additional details, including feature pre-processing, acceleration, and post-hoc ensembling, please refer to [28].

**Remark.** In the tabular domain, years of research into deep and foundation models have culminated in TabPFN v2 [28]—a breakthrough that, for the first time, enables deep models to consistently outperform traditional methods without fine-tuning. However, due to venue constraints, many technical details were omitted from the main paper. For example, the use of randomized tokens was documented in the supplementary material and code. In light of this, we aim to systematically analyze TabPFN v2, as we believe such a study is more impactful than proposing yet another architecture.

# 4 Comprehensive Evaluation of TabPFN v2

Before presenting our core studies, we first extend TabPFN v2's evaluation beyond the original set of datasets to over 300, covering a much broader range of domains, attributes, scales, dimensionalities, and tasks [23, 49, 75, 60], aiming to more thoroughly assess its generalizability and limitations.

## 4.1 Setups

We first adopt the benchmark from [75], comprising 120 binary classification, 80 multi-class classification, and 100 regression tasks. It resolves common issues such as mislabeled data and redundancies from overlapping dataset versions [42], enabling more reliable evaluations. Out of the 300 datasets, 27 belong to the validation set used for checkpoint selection in TabPFN v2 [28]. To avoid evaluation bias, we exclude these datasets and report results on the remaining 273 datasets.

Following the protocol in [21, 22], each dataset is randomly split into training, validation, and test partitions in a 64%/16%/20% ratio. TabPFN v2 predicts test set labels directly using in-context learning, without any additional parameter or hyperparameter tuning. Baseline tabular methods—both deep and traditional—perform hyperparameter tuning using Optuna [1], with 100 trials on the training set and early stopping based on validation performance.

All methods are evaluated using 15 random seeds, and we report the average performance across seeds. For classification tasks, we report accuracy (higher is better), while regression tasks are evaluated using Root Mean Square Error (RMSE; lower is better). For tasks with more than 10 classes, we adopt the built-in Error-Correcting Output Codes (ECOC) strategy for TabPFN v2.

## 4.2 Empirical Results: Strengths of TabPFN v2

We compare TabPFN v2 against 26 representative tabular methods (see Appendix for full references). To assess statistical significance, we apply the Wilcoxon-Holm test with a significance level of 0.05 [14]. As shown in the critical difference diagram in Figure 1 (right), TabPFN v2 consistently outperforms both tree-based methods, such as CatBoost [56], and deep tabular models, including RealMLP [29], ModernNCA [76], TabM [19], TabR [21], and FT-Transformer [22].

To further assess performance, we report the Probability of Achieving the Maximum Accuracy or Minimum RMSE (PAMA) [13], which measures the proportion of datasets on which a method achieves the best performance. As shown in Figure 2, TabPFN v2 attains the highest score, delivering the top results on 26.02% of the datasets—outperforming other methods such as ModernNCA (11.79%) and TabM (9.78%). These results underscore TabPFN v2's strong generalizability.

### 4.3  Empirical Results: Limitations of TabPFN v2

The above evaluation focuses on small- to medium-scale datasets, specifically those with fewer than 10,000 examples. However, as noted in [28], the computational complexity of transformers constrains TabPFN v2's ability to scale effectively to datasets with larger sample sizes or higher dimensionality.

To verify this, we conduct additional evaluations on 18 high-dimensional datasets with $d \geq 2,000$ [36] and 18 large-scale datasets where $N \times d > 1,000,000$. For high-dimensional datasets, we follow the same protocol as before. For large-scale datasets, due to the prohibitive cost of hyperparameter tuning, default hyperparameters are used for all methods. The average ranks of several representative methods are summarized in Table 1. The full results—along with our extensions—are in Section 7.

As shown, TabPFN v2's performance degrades on both large-scale and high-dimensional datasets. On large-scale datasets, it ranks below both CatBoost and RealMLP; on high-dimensional datasets, it even falls behind the simple Logistic Regression (LR) model. Beyond these two limitations, Table 1 also reports results on the 12 datasets in Section 4.2 that contain more than 10 categories, where the ECOC strategy currently used by TabPFN v2 appears ineffective in achieving high accuracy. While increased computational complexity may contribute to this reduced effectiveness, we hypothesize that TabPFN v2 was pre-trained exclusively on small- to medium-scale synthetic datasets with fewer than 10 categories, leading to a mismatch when applied to larger or more complex real-world data.

These results underscore the limitations of TabPFN v2, suggesting areas for further improvement.

## 5  How Does TabPFN v2 Effectively Handle Data Heterogeneity?

Section 4 demonstrates TabPFN v2's excellent generalizability to heterogeneous downstream tasks while also highlighting its current limitations. In the rest of the paper, we first examine the mechanisms behind its strengths, followed by methods to overcome its limitations.

### 5.1  Diving into TabPFN v2's Mechanisms for Heterogeneous Input

**Revisiting the problem.** As noted in Sections 2 and 3, tabular datasets often differ in both the number of attributes (*i.e.*, $d$) and the semantics of those attributes. Even when dimensionalities match, the dimensional semantics from different datasets are typically not directly comparable. A robust tabular foundation model must therefore handle such heterogeneity effectively, enabling it to learn from diverse pre-training datasets and transfer its capabilities to new downstream tasks.

**Tokenization as a feasible solution.** Among prior approaches, the most relevant to TabPFN v2 are token-based methods [64, 22, 74]. The core idea is to convert a $d$-dimensional instance $\boldsymbol{x} \in \mathbb{R}^d$ into a set of $d$ *fixed*-dimensional tokens (each of dimension $k$), with one token per attribute. This enables the use of transformer architectures, which naturally accommodate variability in $d$ across datasets.

To embed each attribute into a shared $k$-dimensional space, prior work either uses pre-defined semantic embeddings [74] (*e.g.*, word vectors of attribute names) or learns dataset-specific embeddings [64, 22]. Given $d$ attribute-specific tokens $[\boldsymbol{r}_1, \ldots, \boldsymbol{r}_d] \in \mathbb{R}^{d \times k}$, each instance $\boldsymbol{x}_i \in \mathbb{R}^d$ can then be transformed into $[x_i^1 \cdot \boldsymbol{r}_1, \ldots, x_i^d \cdot \boldsymbol{r}_d] \in \mathbb{R}^{d \times k}$, where $x_i^j$ denotes the $j$-th element of $\boldsymbol{x}_i$.

By embedding all attributes into a shared, fixed-dimensional feature space, this approach allows the transformer to learn transferable patterns and knowledge from heterogeneous datasets.

**Difficulty in direct generalization.** While appealing, the aforementioned methods face a notable challenge when applied to downstream tasks: attribute names or semantics are not always accessible, as discussed in Section 2. Although it is possible to learn dataset-specific attribute tokens, doing so incurs additional computational cost and prohibits the reuse of previously learned tokens. Consequently, this limits the direct generalization of the foundation model to new tasks.

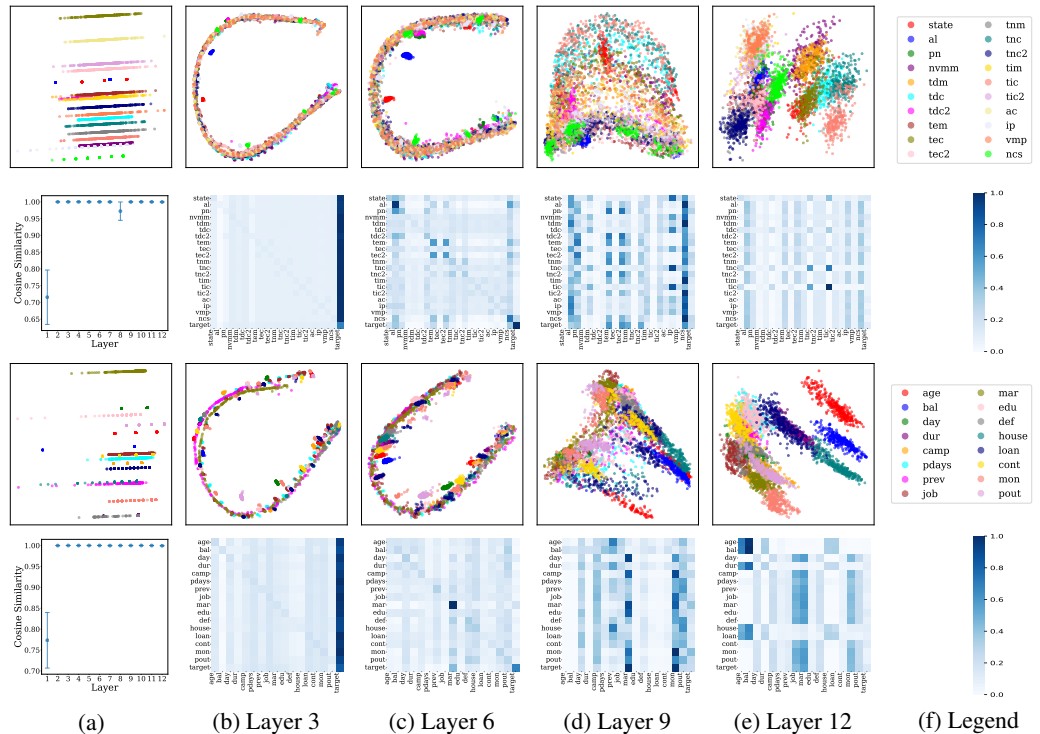

Figure 3: Attribute relationships inferred by TabPFN v2. The first and third rows show PCA projections of the $d$ attribute tokens from all $N$ training instances at various layers for the *churn* and *bank* datasets. Colors indicate different attributes (see legend on the right). The second and fourth rows display the attribute-wise attention maps. Each matrix cell represents the average attention weight between attributes; *the last element along each axis (*e.g.*, the last column and row) corresponds to the label.* The first plots in the second and fourth rows summarize the cosine similarity of attention maps across random seeds. See text for details.

**TabPFN v2's mechanisms.** TabPFN v2 builds on prior token-based methods by representing each instance $\boldsymbol{x}_i$ as a sequence of tokens. However, rather than assigning a deterministic token to each attribute, TabPFN v2 samples random tokens at inference time. Specifically, it learns a shared vector $\boldsymbol{u} \in \mathbb{R}^k$ that lifts each element of $\boldsymbol{x}_i$ into a $k$-dimensional space. To distinguish attributes, TabPFN v2 adds a random perturbation to each one.[2] For the $j$-th attribute (*i.e.*, $x_i^j$), the representation becomes $x_i^j \cdot \boldsymbol{u} + \boldsymbol{r}_j$, where $\boldsymbol{r}_j = \boldsymbol{W}\boldsymbol{p}_j$. Here, $\boldsymbol{p}_j \in \mathbb{R}^{k'}$ is a randomly generated vector, and $\boldsymbol{W} \in \mathbb{R}^{k \times k'}$ is a learned projection matrix that conditions the perturbation. The full instance $\boldsymbol{x}_i$ is then represented as:

$$[x_i^1 \cdot \boldsymbol{u} + \boldsymbol{r}_1, \dots, x_i^d \cdot \boldsymbol{u} + \boldsymbol{r}_d, \tilde{\boldsymbol{y}}_i] \in \mathbb{R}^{k \times (d+1)}, \tag{1}$$

where the last token $\tilde{\boldsymbol{y}}_i$ encodes the label information (see Figure 1 for an illustration).

### 5.2 TabPFN v2 Internalizes Attribute Token Learning

TabPFN v2's randomized tokenization scheme eliminates the need to define attribute- or dataset-specific tokens across tasks, thereby *syntactically* enabling direct application of the pre-trained model. At first glance, this may appear to disregard the valuable *semantic* meaning of attributes. However, we show that through in-context learning, TabPFN v2 can consistently infer relationships among attributes within a dataset—despite the randomness introduced during tokenization. Specifically, we analyze the behavior of attribute tokens from three perspectives, as illustrated in Figure 3, using two representative downstream datasets: *churn* and *bank*.

First, we visualize the attribute token embeddings (*i.e.*, the first $d$ tokens in Equation (1)) across all $N$ training instances. The first and third rows of Figure 3 present PCA projections of these $N \times d$ tokens at the input stage and after transformer layers $\{3, 6, 9, 12\}$, with colors indicating different attributes. Initially, tokens from different attributes appear randomly scattered. However, as the input progresses through the transformer layers, these tokens become increasingly structured. For example,

---

[2]This detail was identified from the supplementary material and code of [28].

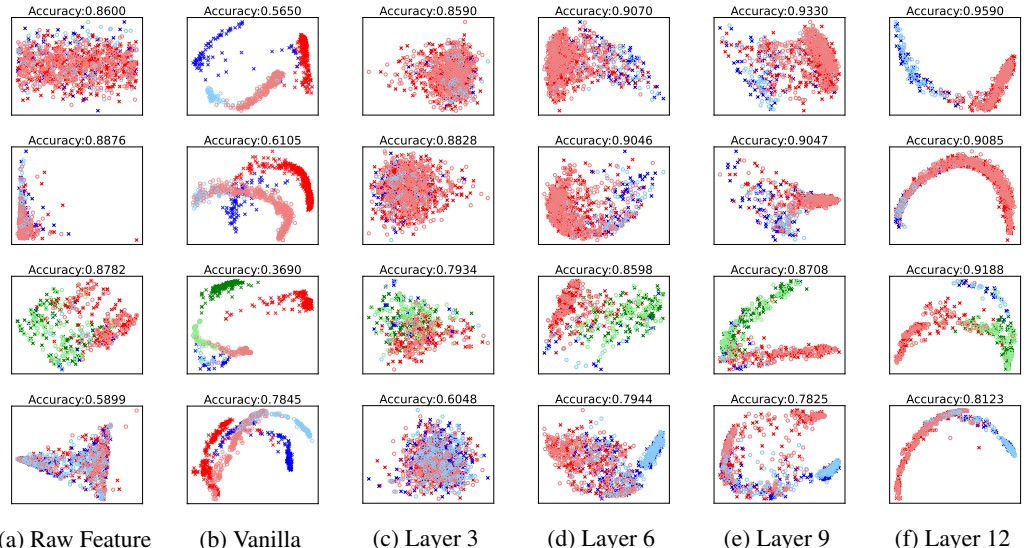

| (a) Raw Feature | (b) Vanilla | (c) Layer 3 | (d) Layer 6 | (e) Layer 9 | (f) Layer 12 |

Figure 4: Visualization of the extracted instance features from four datasets: *churn* (first row, binary), *bank* (second row, binary), *website_phishing* (third row, three classes), and *KDD* (fourth row, binary). Blue and red indicate classes; darker crosses and lighter circles denote training and test samples. (a) shows the raw input features (*e.g.*, $x_i$), while (b) presents embeddings from the vanilla strategy. (c)-(f) display embeddings produced by our method at different layers. Classification accuracy is reported by training a linear logistic regression model on the training embeddings and evaluating on the test set.

in the *bank* dataset (which predicts term deposit subscriptions), attributes such as "job," "education," and "balance" eventually cluster into semantically coherent groups.

Second, we examine attribute-wise attention patterns across layers, including attention to the label token. The second and fourth rows of Figure 3 show heatmaps of attention weights averaged over heads and training instances. Each row in the heatmap represents the attention distribution from one attribute to all others; *the last element along each axis (*e.g.*, the last column and row) corresponds to the label.* Darker shades indicate stronger attention. We observe a consistent pattern across datasets: in early layers, attributes predominantly attend to the label token, likely to absorb task-specific signals. In intermediate layers, attention becomes more uniformly distributed, facilitating information exchange across attributes. In deeper layers, attention concentrates on semantically relevant attributes, suggesting the model has inferred inter-attribute relationships useful for prediction.

Lastly, to assess robustness against random token initialization, we compute attribute-wise attention weights across 10 runs. The cosine similarities and variances of these attention patterns are summarized in the first plots of the second and fourth rows in Figure 3. The results confirm that attention patterns remain stable across runs, except for the first layer.

**Remark.** The above results suggest that TabPFN v2 can reliably infer meaningful attribute relationships through in-context learning. Although input embeddings are randomized, they consistently differentiate attributes across instances—functionally akin to one-hot encodings. Pre-training on diverse tasks thus enables the model to extract predictive patterns (*e.g.*, co-occurrence across attributes, value distributions, and relative magnitudes) directly from the statistical structure of each dataset, without relying on pre-defined attribute semantics. As a result, the model effectively internalizes attribute token learning within the inference process. See the Appendix for further discussion.

# 6 TabPFN v2 Can Be Transformed into an Effective Feature Encoder

In Section 5, we show that TabPFN v2's in-context learning process infers meaningful attribute relationships. Here, we examine whether TabPFN v2 also produces separable instance representations.

## 6.1 Naive Feature Extraction Fails

As shown in Figure 1 (left), TabPFN v2 makes predictions based on the output token corresponding to the (dummy) label embedding $\tilde{y}^*$ of the test instance. This output token can thus be interpreted as the

instance embedding for the test example. A natural extension to obtain embeddings for the training instances is to extract the output tokens corresponding to the training label embeddings $\{\tilde{\boldsymbol{y}}_i\}_{i=1}^{N}$.

However, as shown in Figure 4 (b), this naive approach leads to surprisingly discrepant feature distributions between training (darker cross) and test (lighter circle) examples. As a result, a linear classifier trained on these embeddings performs poorly on the test set. We attribute this discrepancy to the distinct roles of labeled training data and unlabeled test data in TabPFN v2's in-context learning process. Specifically, the label embeddings for the training instances are derived from true labels, whereas those for the test instances rely on dummy labels. This mismatch renders the resulting output embeddings *non-comparable* between training and test instances.

## 6.2 Leave-one-fold-out Feature Extraction

To address this challenge, we propose a leave-one-fold-out strategy that enables the extraction of comparable embeddings for training and test data. In the TabPFN v2 framework, we treat examples with true labels as the support set $\mathcal{S}$, and those with dummy labels as the query set $\mathcal{Q}$. Under the standard configuration, $\mathcal{S}$ corresponds to the labeled training set and $\mathcal{Q}$ to the unlabeled test instances. To extract comparable embeddings for the training examples, they must also be included in $\mathcal{Q}$ with dummy label embeddings. This, however, creates a dilemma: effective in-context learning relies on maximizing the size of $\mathcal{S}$ to ensure sufficient knowledge transfer to $\mathcal{Q}$. Including training examples in $\mathcal{Q}$ thus competes with the need to keep $\mathcal{S}$ as large as possible.

To overcome this dilemma, we partition the training set into multiple folds (*e.g.*, 10). In each round, one fold serves as $\mathcal{Q}$—with dummy labels used for embedding extraction—while the remaining folds form $\mathcal{S}$ with true labels. This setup preserves sufficient label supervision in $\mathcal{S}$ while enabling the extraction of embeddings for training instances in $\mathcal{Q}$. Results in Figure 4 (c)-(f) show that embeddings extracted by this strategy (with 10 folds) more faithfully capture dataset structure. We observe that TabPFN v2 simplifies the original tabular data distributions, transforming datasets into nearly linearly separable embedding spaces—especially after intermediate transformer layers.

We also experimented with a variant that uses the same context and query without partitioning, where the context contains all training samples and the query set includes both training and test points. This "non-partitioned" strategy improves upon the vanilla feature extraction baseline but still underperforms compared to our proposed leave-one-fold-out method. We attribute this to role ambiguity: query points that appear in the support set (either with dummy or ground-truth labels) are treated inconsistently, preventing the network from fully distinguishing between training and test roles and thereby degrading feature consistency. Detailed results for this variant are provided in the Appendix.

## 6.3 Validation of Embedding Quality

To validate the quality of the extracted embeddings, we train a logistic regression on embeddings derived from the training set and evaluate it on test set embeddings. The average rank across 29 classification datasets from the tiny benchmark2 in [75] is reported in Table 2.

Table 2: Average rank (lower is better) of TabPFN v2 and linear classifiers trained on the extracted embeddings across 29 classification datasets. Combined: embeddings from up to three layers (from the 12 available layers) are selected and concatenated, based on the validation set performance.

| ↓ | TabPFN v2 | Vanilla | Layer 6 | Layer 9 | Layer 12 | Combined |
|---|---|---|---|---|---|---|
| Rank | 2.69 | 5.97 | 4.28 | 4.00 | 2.12 | **1.94** |

Remarkably, training a simple linear classifier on the extracted embeddings achieves performance comparable to that of TabPFN v2's in-context learner. Furthermore, concatenating embeddings from multiple layers (*e.g.*, both output and intermediate representations) can sometimes lead to even better results. These findings underscore TabPFN v2's potential as a strong and versatile feature encoder, suggesting broader applicability in downstream tasks such as tabular data analysis.

# 7 Improving TabPFN v2 via Test-Time Divide-and-Conquer

This section addresses the limitations discussed in Section 4.3, aiming to extend TabPFN v2's applicability beyond the boundaries. Specifically, we propose post-hoc divide-and-conquer strategies inspired by Chain-of-Thought (CoT) prompting [70], which decompose challenging tasks into simpler subtasks that TabPFN v2 can effectively handle.

## 7.1 High Dimension Datasets

High-dimensional datasets [36] present a unique challenge due to the quadratic complexity of TabPFN v2 with respect to the number of dimensions. To mitigate this, we propose subsampling the feature space into smaller subsets, processing each subset independently, and combining the predictions in an ensemble (bagging) fashion, similar to random forests [8].

In detail, we iteratively sample $m$ subsets, each containing $d' < d$ randomly selected attributes. For each subset, we leverage TabPFN v2's ability to handle lower-dimensional data to obtain predictions. We denote this divide-and-conquer and then ensemble strategy as TabPFN v2*, which aggregates outputs using averaging (for regression) or majority voting (for classification). To address high-dimensional tasks, we introduce a baseline variant, TabPFN v2-PCA, which incorporates dimensionality reduction. Specifically, TabPFN v2-PCA reduces the feature dimension to 500 using PCA to satisfy the input constraints of TabPFN v2. This process is repeated multiple times with different PCA projections, and the resulting predictions are aggregated via bagging to improve robustness.

Figure 5 (left) summarizes the results on 18 high-dimensional *classification* datasets. A variant that utilizes PCA to reduce the dimensionality, together with bagging, resolves the dimensionality issue to some extent. TabPFN v2* with $d' = 500$ and $m = \lceil d/d' \rceil$ significantly increases the mean accuracy (to the highest), effectively extending TabPFN v2's scalability to datasets with $d \geq 2000$.

## 7.2 Multi-Class Problems with More Than 10 Classes

To extend TabPFN v2 to tasks with more than 10 categories, we propose a decimal encoding approach that decomposes multi-class problems into multiple 10-class subproblems, ensuring compatibility with TabPFN v2's constraints.

For a task with $C > 10$ classes, we encode each label $y \in [C]$ as a $t$-digit decimal representation, where $t = \lceil \log_{10} C \rceil$. For each digit position $j \in \{1, \ldots, t\}$, we train a separate TabPFN v2 model $f_j$ to predict the $j$-th digit. During inference, the predicted digits are reconstructed to obtain the final class label. This strategy is also developed in [48], and we denote it as TabPFN v2-DPT. As the decimal encoding inherently introduces artificial correlations among classes — classes that share the same digit at a given position are grouped together, even if they are semantically unrelated. To mitigate this effect, our TabPFN v2* randomly permutes the class-to-digit mapping $\sqrt{C}$ times, leading to different groupings in each run, and the prediction results are ensembles to improve robustness.

We consider the following variants of TabPFN v2 to address the 10-class limit in classification tasks:

- TabPFN v2-ECOC: We use the implementation provided in the official TabPFN extensions repository, which applies Error-Correcting Output Codes (ECOC).
- TabPFN v2-DPT: We encode each class label as a $t$-digit decimal string and train a separate TabPFN v2 to predict each digit. For instance, a 15-class problem is decomposed into two subproblems: one for the tens digit (classes $\{0, 1\}$) and one for the ones digit (classes $\{0, \ldots, 9\}$). The predicted digits are then decoded to recover the final class label.

We implement TabPFN v2 * based on TabPFN v2-DPT for efficiency. Specifically, TabPFN v2 * permutes the class-to-digit mapping $\sqrt{C}$ times. For fair comparison, we also increase the number of ensembles in TabPFN v2-DPT to $\sqrt{C}$ per digit to match the total number of predictions. As shown in Figure 5 (middle), this approach achieves the second-best mean accuracy on 12 datasets with more than 10 classes while preserving computational efficiency.

## 7.3 Large-Scale Datasets

For large-scale datasets, we randomly sample 10,000 training examples from the full training set as the support set and treat the remaining training examples and test instances as the query set. We extract their embeddings to form a new tabular dataset, on which a logistic regression classifier is trained to make predictions on the test set embeddings. This process is repeated four times, and the final predictions are aggregated. We denote this version as TabPFN v2*-SQ.

We also investigate integrating TabPFN v2 with decision trees to handle large-scale tasks. We note that a similar strategy was mentioned in [28] to handle within-dataset heterogeneity for a drastically different purpose. Specifically, we sample 32 subsets from the original training set, each containing

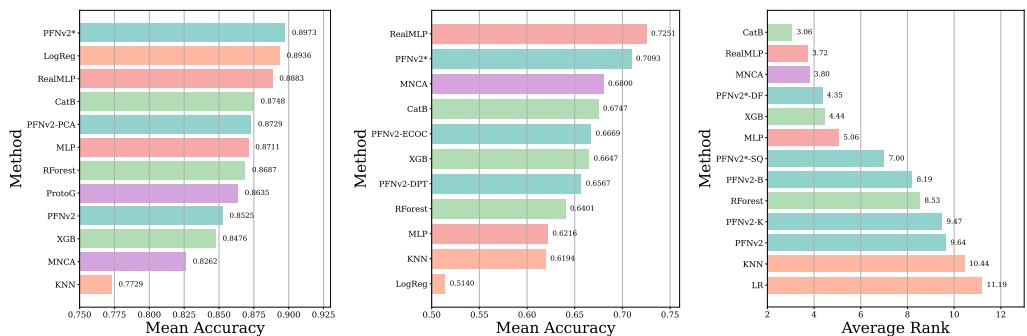

Figure 5: "*" indicates our extension. **Left**: Mean accuracy on 18 high-dimensional datasets. "-PCA" is another variant using PCA to reduce dimensions. **Middle**: Mean accuracy on 12 datasets with more than 10 classes. "-ECOC" denotes the multi-class ECOC strategy implemented by [28]. **Right**: Average rank on 18 large-scale datasets. "-B" refers to the variant that randomly subsamples 10,000 training examples four times and aggregates their predictions. "-K" denotes the variant that selects a representative subset of 10,000 training examples based on proximity to prototypes obtained via KMeans. All variants improve TabPFN v2.

60% of the original data (sampled without replacement). For each subset, we first train a shallow decision tree by setting the minimum number of samples required to split an internal node to 10,000. The decision tree partitions the training set into smaller, more manageable subsets. During inference, a test instance is first passed through each of the shallow decision tree to a leaf node and then predicted by the corresponding TabPFN v2 model. The predictions from all 32 models are aggregated. We denote this extension as TabPFN v2*-DF.

We consider the following variants of TabPFN v2 to scale to larger datasets:

- TabPFN v2-DT: A shallow decision tree is trained with a minimum split size of 10,000. The tree partitions the dataset into smaller subsets, and a separate TabPFN v2 model is applied to each leaf node. At inference, a test instance is routed through the tree to a corresponding leaf, where it is predicted by the respective TabPFN v2 model.
- TabPFN v2-B: A bagging-based variant that randomly samples 10,000 training examples four times and aggregates their predictions.
- TabPFN v2-K: Selects a representative subset of 10,000 training examples based on proximity to KMeans-derived prototypes.

Our TabPFN v2 *-DF is an extension of TabPFN v2-DT to a forest-based ensemble. Specifically, we sample 32 subsets from the original training data, each containing 60% of the samples (without replacement), and train a separate TabPFN v2*-DT model on each subset. During inference, predictions from all 32 models are aggregated—*e.g.*, by majority voting or averaging—depending on the task type.

Figure 5 (right) shows the average rank results, including TabPFN v2*-SQ and TabPFN v2*-DF alongside variants using bagging and KMeans-based sampling. We observe that all variants improve upon the vanilla TabPFN v2 on large-scale datasets, with TabPFN v2*-DF and TabPFN v2*-SQ achieving the most significant improvement.

# 8  Conclusion

We present a timely investigation into TabPFN v2, a groundbreaking foundation model for tabular tasks. Our analysis uncovers the core mechanism behind TabPFN v2's strong performance across heterogeneous tabular datasets: it can infer attribute relationships on-the-fly—even from randomly initialized token inputs—without relying on pre-defined semantics or learning dataset-specific representations. We also demonstrate that TabPFN v2 can be repurposed as a powerful feature encoder, enabling broader applications such as data visualization and diagnostic analysis. To address its limitations in more complex data regimes, we introduce post-hoc divide-and-conquer strategies that extend TabPFN v2's utility without requiring model re-training. Together, these contributions offer fresh insights into advancing the development and application of foundation models for tabular data.

## Acknowledgment

This work is partially supported by National Key R&D Program of China (2024YFE0202800), Fundamental Research Funds for the Central Universities (2024300373), Collaborative Innovation Center of Novel Software Technology and Industrialization.

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

## A    Additional Related Work

**Learning with Tabular Data**. Tabular data is prevalent across diverse fields, including healthcare, finance, and education [43, 32, 58, 2]. Tree-based models, such as XGBoost [12], LightGBM [38], and CatBoost [56], have long dominated this domain. However, recent advances in deep neural networks (DNNs) have demonstrated strong potential for tabular data [7]. Popular architectures like multi-layer perceptrons [22, 37] and Transformers [31] have been adapted to tabular tasks, alongside custom architectures designed specifically for tabular data [41, 68].

Deep tabular methods can be broadly categorized into two types. The first type directly processes raw features [29, 21, 76], sometimes incorporating feature-specific encoding strategies [20]. The second type tokenizes features, transforming an example into a set of tokens [64, 31, 59]. Comprehensive benchmarks have been developed to evaluate these methods across diverse datasets [23, 49, 75, 60], highlighting the strengths and weaknesses of deep tabular models in various scenarios.

**Variants of TabPFN.** TabPFN's success stems from its pre-training on massive synthetic datasets, enabling strong in-context learning performance on small-scale classification tasks [27]. Motivated by its capabilities, researchers have explored a variety of applications, including tabular data generation [47], anomaly detection [61], and time series forecasting [30]. [53] provided a bias-variance analysis of TabPFN, offering insight into its generalization behavior. Another line of research focuses on improving scalability by addressing TabPFN's sensitivity to context size [16, 73]. Further strategies to enhance downstream performance include context adaptation with nearest neighbor [65], partial fine-tuning [17, 45], pre-training on real-world datasets [48], scalable ensemble [46], and more powerful and efficient pre-training on synthetic data [57]. Most of these variants remain restricted to classification tasks due to limitations in TabPFN v1.

The recently introduced TabPFN v2 [28] extends TabPFN to support regression tasks and accommodate larger context sizes. In this paper, we conduct a comprehensive evaluation of TabPFN v2, analyze its strengths, and introduce methods to overcome its scalability and applicability challenges.

## B    Evaluation Details

**Experimental Compute Resources.** All experiments were conducted using 4 NVIDIA RTX 6000 Ada GPUs and 2 Intel(R) Xeon(R) Platinum 8352V CPUs.

Please refer [75] for details of the 300 small to medium datasets. For high-dimensional datasets, we selected 18 datasets with more than 2000 features from the scikit-feature repository. Detailed statistics of high-dimensional datasets and large-scale datasets are reported in Table 3 and Table 4.

We follow [75] and use different colors to represent various categories of methods in the result figures, ensuring clarity and easy comparison. In Figure 1 (right) and Figure 2 of the main paper, we compare the following methods:

Table 3: Dataset Information for High-Dimensional Data Experiments: A collection of 18 datasets with varying numbers of instances, features, and classes used in our high-dimensional experiments.

| Dataset | #Instances | #Features | #Classes | Dataset | #Instances | #Features | #Classes |
|---|---|---|---|---|---|---|---|
| BASEHOCK | 1993 | 4862 | 2 | lung | 203 | 3312 | 5 |
| PCMAC | 1943 | 3289 | 2 | warpPIE10P | 210 | 2420 | 10 |
| RELATHE | 1427 | 4322 | 2 | orlraws10P | 100 | 10304 | 10 |
| ALLAML | 72 | 7129 | 2 | Prostate_GE | 102 | 5966 | 2 |
| CLL_SUB_111 | 111 | 11340 | 3 | SMK_CAN_187 | 187 | 19993 | 2 |
| colon | 62 | 2000 | 2 | warpAR10P | 130 | 2400 | 10 |
| GLI_85 | 85 | 22283 | 2 | arcene | 200 | 10000 | 2 |
| GLIOMA | 50 | 4434 | 4 | gisette | 7000 | 5000 | 2 |
| leukemia | 72 | 7070 | 2 | TOX_171 | 171 | 5748 | 4 |

Table 4: Dataset Information for Large-scale Data Experiments.

| Dataset | #Instances | #Features | #Classes | Dataset | #Instances | #Features | #Classes |
|---|---|---|---|---|---|---|---|
| BNG(credit-a) | 1,000,000 | 15 | 2 | CDC_Indicators | 253,680 | 21 | 2 |
| Higgs | 1,000,000 | 28 | 2 | Smoking_signal | 991,346 | 23 | 2 |
| nomao | 34,465 | 118 | 2 | sf-police-incidents | 2,215,023 | 8 | 2 |
| Data_Crowdfunding | 671,025 | 11 | 4 | Fashion-MNIST | 70,000 | 784 | 10 |
| covertype | 581,012 | 54 | 7 | jannis | 83,733 | 54 | 4 |
| poker-hand | 1,025,009 | 10 | 10 | volkert | 58,310 | 180 | 10 |
| Airlines_DepDelay | 10,000,000 | 9 | - | Wave_Energy_Farm | 36,043 | 99 | - |
| UJIndoorLoc | 21,048 | 520 | - | blogfeedback | 60,021 | 276 | - |
| microsoft | 1,200,192 | 136 | - | yahoo | 709,877 | 699 | - |

- **Classical Methods** (■): The classical methods include Dummy, Logistic Regression (LR), K-Nearest Neighbors (KNN), Support Vector Machines (SVM), Naive Bayes, Linear Regression, and DNNR [52], which serve as basic baselines for classification and regression tasks.
- **Tree-based Methods** (■): Tree-based methods such as Random Forest [8], XGBoost [12], Light-GBM [38], and CatBoost [56] are known for their high performance on tabular data.
- **MLP variants** (■): MLP variants, including vanilla MLP, MLP-PLR, Self-Normalizing Neural Networks [41], Residual Network [22], and TabM [19] enhance the flexibility and generalization of traditional MLP architectures through advanced regularization and residual connections.
- **Special Architectures** (■): Methods with specially designed architectures, such as DCNv2 [68], DANets [10], and TabCaps [9], focus on improving feature interaction and abstraction to capture complex relationships in tabular data.
- **Token-based Methods** (■): Token-based methods like AutoInt [64], TabTransformer [31], FT-Transformer [22], and ExcelFormer [11] represent features as tokens, enabling models to capture higher-order interactions through attention mechanisms.
- **Regularization-based Methods** (■): Regularization-based methods, including TANGOS [35], SwitchTab [72], and PTaRL [78], aim to improve model generalization by incorporating regularization techniques during training to enhance the robustness of predictions.
- **Tree-mimic Methods** (■): Tree-mimic methods, such as NODE [55], GrowNet [5], and TabNet [4], combine the interpretability of decision trees with the power of deep learning, employing attention mechanisms to select important features.
- **Context-based Methods** (■): Context-based methods like TabR [21] and ModernNCA [76] leverage contextual information from the training data to improve predictions by utilizing neighborhood-based and in-context learning strategies.

In addition to the aforementioned methods, for other experimental results, we will demonstrate the performance of **TabPFN v2 and its variants**, which are represented by emerald teal (■), ensuring that their experimental effects are clearly distinguished from the other methods.

*Remark*. The standard checkpoint released by TabPFN employs a feature grouping size of 2, which complicates the analysis of individual feature embeddings and inter-feature relationships. To facilitate such analysis, we use a modified checkpoint with *group size=1* for the experiments in Figure 3 of the main paper, which is available at HuggingFace.

# C   Additional Feature Extraction Results

In this section, we provide further results on regression tasks and additional variants of our feature extraction strategy to validate the effectiveness and robustness of the proposed leave-one-fold-out method with TabPFN v2. We also compare this supervised approach to two unsupervised embedding extraction methods, as well as a non-partitioned variant that uses the same context and query without data folding.

## C.1   Additional results for leave-one-fold-out feature extraction strategy with TabPFN v2 on regression tasks

To further demonstrate the versatility of our approach, we evaluate TabPFN v2 on regression tasks by extracting embeddings using the leave-one-fold-out strategy and training simple linear regressors on top. As shown in Table 5, our embeddings consistently outperform both vanilla and raw features across different regressors, achieving the best average rank. This result indicates that TabPFN v2 can also serve as an effective feature extractor for regression problems, further supporting its general applicability beyond classification tasks.

Table 5: Average rank comparison of embeddings on regression tasks using different regressors. Lower is better. LR denotes Linear Regression.

| Extraction Strategy | Vanilla | Vanilla | Raw | Raw | Ours | Ours |
| Regressor | LR | RidgeCV | LR | RidgeCV | LR | RidgeCV |
| --- | --- | --- | --- | --- | --- | --- |
| Avg. Rank | 5.58 | 5.08 | 3.83 | 3.17 | **2.08** | **1.25** |

## C.2   Comparison Between Supervised and Unsupervised Feature Extraction

Our leave-one-fold-out strategy explicitly incorporates label information and accounts for the distinct roles of training and test instances. To further understand the nature of the extracted embeddings, we compare this supervised strategy to two unsupervised alternatives provided by the TabPFN extensions repository.

Let $X \in \mathbb{R}^{n \times d}$ denote a dataset with $n$ samples and $d$ features:

- **Unsupervised-Dummy**: Each sample is paired with a constant pseudo-target $\mathbf{y} = \mathbf{0} \in \mathbb{R}^n$, forming a regression task. The embedding $E_{\mathrm{D}} \in \mathbb{R}^{n \times h}$ is obtained by extracting the output tokens of the TabPFN regressor.
- **Unsupervised-Permute**: For each feature $j \in \{1, \ldots, d\}$, we treat $\mathbf{x}^{(j)} = X_{:,j}$ as a pseudo-target and use the remaining features $X^{(-j)}$ as input. Depending on the type of $\mathbf{x}^{(j)}$, TabPFN is applied in classification or regression mode to obtain $E^{(j)} \in \mathbb{R}^{n \times h}$. These embeddings are concatenated to a high-dimensional form:
$$E_{\mathrm{P}} = \mathrm{concat}(E^{(1)}, E^{(2)}, \ldots, E^{(d)}) \in \mathbb{R}^{n \times (d \cdot h)}.$$

We compare these unsupervised methods with our supervised leave-one-fold-out strategy in Table 6. Overall, unsupervised approaches underperform compared to the supervised ones and fail to recover the classification ability of TabPFN v2. This performance gap arises because label information is introduced only post hoc via a linear classifier rather than during embedding extraction. Among the unsupervised methods, the permutation-based approach performs better, likely due to its ability to encode attribute-specific structure.

Figure 6 presents a visual comparison of embeddings produced by the three methods using the same color and marker scheme as in Figure 4. Since unsupervised methods lack label supervision during embedding generation, their embeddings tend to scatter broadly without forming well-separated clusters by class. These results further highlight the contrasting goals of supervised and unsupervised strategies—class separation versus feature distribution coverage, respectively.

## C.3   Results for the Non-Partitioned Feature Extraction Variant

Using the same context and query without partitioning, we experimented with a variant where the context contains all training data and the query set includes both training and test points. This

Table 6: Average rank (lower is better) of TabPFN v2 and a linear classifier trained on the extracted embeddings across 29 classification datasets. In addition to the supervised feature extraction strategy considered in the main paper (including the vanilla one, our leave-one-fold-out, and the version based on the combined features), we compare with two *unsupervised* embedding extraction approaches by appending a column of *dummy* labels with zero values and *permuting* each column as labels, respectively.

| ↓ | TabPFN v2 | Vanilla | Dummy | Permute | Ours | Combined |
|------|-----------|---------|-------|---------|------|----------|
| Rank | 2.66 | 5.72 | 4.90 | 3.69 | 2.16 | **1.88** |

(a) Raw Feature     (b) Vanilla     (c) Dummy     (d) Permute     (e) Ours

Figure 6: Comparison between unsupervised and supervised (ours) feature extraction. Visualization of extracted embeddings for four datasets: *churn* (first row, two classes), *bank* (second row, two classes), *KDD* (third row, two classes), and *website_phishing* (fourth row, three classes). We use crosses to denote training examples and circles to denote test examples. (a) shows the raw features, while (b) presents the embeddings extracted using the vanilla strategy. (c) and (d) refer to *unsupervised* embedding extraction approaches by appending a column of dummy labels with zero values and permuting each column as labels, respectively. (e) depicts the embeddings obtained using our proposed methods. The accuracy value is calculated by training a linear model (logistic regression) over the extracted embeddings on the training set and predicting on the test set.

*non-partitioned* strategy improves upon the vanilla feature extraction baseline but still underperforms compared to our proposed leave-one-fold-out method. We attribute this to *role ambiguity*: query points that also appear in the support set (with dummy or ground-truth labels) are treated inconsistently, preventing the network from fully distinguishing between training and test roles and thereby degrading feature consistency.

## D   Influence of Key Modules and Meta-Feature Analysis

We investigate the influence of two key components in TabPFN v2, *i.e.*, the feature engineering that pre-processes the raw features of a given tabular dataset and the post-hoc ensembleing. In addition, we analyze the conditions under which TabPFN v2 performs well or poorly through a meta-feature-based classification analysis.

Table 7: Performance ranking (lower is better) of different feature extraction strategies on classification tasks. The non-partitioned variant uses the same context and query without data folding.

| Method | Avg. Rank |
|---|---|
| Leave-one-fold-out (Ours) | **2.10** |
| TabPFN v2 | 2.72 |
| Non-partitioned (12 layers) | 3.07 |
| Non-partitioned (9 layers) | 4.29 |
| Non-partitioned (6 layers) | 4.43 |
| Vanilla (12 layers) | 5.86 |
| Vanilla (9 layers) | 6.62 |
| Vanilla (6 layers) | 6.90 |

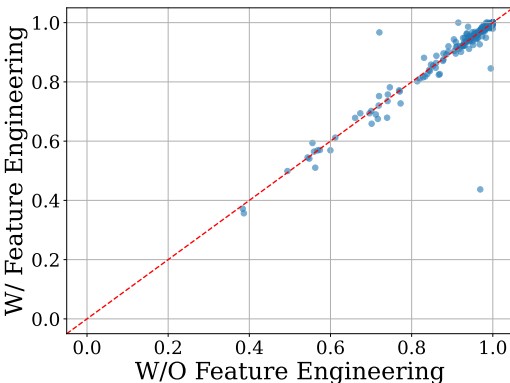

Figure 7: Scatter plot comparing the normalized Accuracy/$R^2$ scores. The x-axis represents the normalized Accuracy/$R^2$ scores without Feature Engineering, while the y-axis represents the normalized Accuracy/$R^2$ scores with Feature Engineering. The red dashed line ($y = x$) serves as a reference, indicating equal performance.

**Feature engineering**. TabPFN v2 pre-processes the features of a given tabular dataset with various strategies, such as quantile, category shuffling, SVD, and power transform. Specifically, we examine the effects of adding fingerprint features (`add_fingerprint_feature`) and polynomial features (`polynomial_features`) to the raw tabular data. The results indicate that TabPFN v2 performs well even without the use of these engineered features, suggesting that, for the benchmark datasets of [75], these specific feature engineering techniques do not provide a significant improvement. This finding highlights the robustness of TabPFN v2 and its ability to handle raw features effectively, without the need for extensive pre-processing or feature construction. We show the influence of this step in Figure 7.

**Model ensemble**. Post hoc ensembling (PHE) involves applying TabPFN v2 to the datasets multiple times with different perturbations and aggregating the predictions of these base models at different temperatures. We show the change of performance of TabPFN v2 w.r.t. the number of ensemble numbers (*i.e.*, the number of base models) in Figure 8. On the benchmark of [75], we observe that, overall, ensemble methods improve performance, with larger ensemble sizes yielding better results. However, we also note that even without ensembling, TabPFN v2 performs exceptionally well, and the relative performance gain from ensembling is limited. This suggests that while ensembling can provide further improvements, the base TabPFN v2 model is already highly effective on its own. The equivariant property described in [3] provides insight into this phenomenon. Since TabPFN v2 introduces random tokens to handle heterogeneous features, the model becomes less sensitive to the arbitrary ordering of features, effectively enforcing equivariance in this aspect. As a result, the benefits of ensembling through feature order permutations are less pronounced compared to TabPFN v1.

**Meta-Feature Analysis of TabPFN v2 Performance.** To better understand the conditions under which TabPFN v2 performs well or poorly, we conducted a meta-learning-based classification analysis using 300 datasets. Specifically, we used the average rank of TabPFN v2 across datasets as a performance indicator. The threshold for classification was set at the mean rank, 6.31. Datasets where

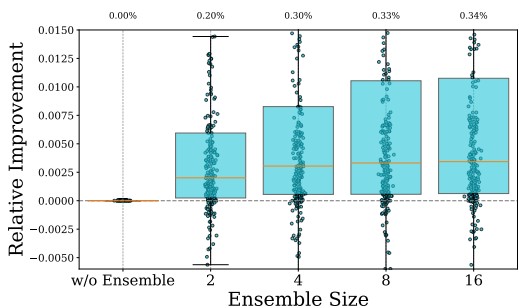

Figure 8: Box plot of relative performance improvements of TabPFN v2 with post hoc ensembling (PHE) across different ensemble sizes (2, 4, 8, and 16 base models). The relative improvement is calculated as the performance gain over the non-ensemble model, where higher values indicate stronger performance. The box plots show the median, interquartile range (IQR), and outliers for each ensemble size.

TabPFN v2 achieved a rank lower than or equal to 6.31 were labeled as "Good", while those with a higher rank were labeled as "Bad". We extracted meta-features from each dataset and used them to train a decision tree classifier, aiming to distinguish between the "Good" and "Bad" categories. All the meta-features utilized in this task are detailed in Table 8, accompanied by their respective explanations. A visual depiction of a simple depth-3 decision tree is shown in Figure 9, and the tree reveals key factors that influence the effectiveness of TabPFN v2. The decision tree visualizes how to predict whether TabPFN v2 performs well ("Good") or poorly ("Bad") on a given dataset, based on dataset meta-features: The root node splits on the number of instances (`nr_inst`), indicating that TabPFN v2 tends to perform better on datasets with fewer than 24,350 samples. For these smaller datasets, the left subtree further splits on the mean joint entropy (`joint_ent.mean`), where higher values (greater than 3.028) are associated with improved performance. For datasets with lower mean joint entropy ($\leq$ 3.028), TabPFN v2 also tends to perform well when the number of rows is relatively small ($\leq$ 4,862). In contrast, the right subtree, which represents larger datasets, reveals that a low standard deviation of the interquartile range (`iq_range.std`) across features ($\leq$ 0.657) is linked to poorer model performance.

# E Detailed Results

We list the detailed results of TabPFN v2 and our extensions on various benchmarks.

- We present the **main results of TabPFN v2** on 300 datasets in Table 9. The table includes accuracy for classification tasks and RMSE (Root Mean Squared Error) for regression tasks, along with the corresponding mean and standard deviation for each dataset. Notably, we excluded 27 datasets from these results in Table 9, as they were used by TabPFN v2 to select the best checkpoint. These excluded datasets, which are not shown in Figure 1 (right) and Figure 2 of the main paper, include:

  (1) ada_prior, allbp, baseball, delta_ailerons, eye_movements, eye_movements_bin, GAMETES_Epistasis_2-Way_20atts_0.1H_EDM-1_1, hill-valley, JapaneseVowels, jungle_chess_2pcs_raw_endgame_complete, led24, longitudinal-survey, page-blocks, ringnorm, rl, thyroid-ann, waveform-5000,
  (2) debutanizer, delta_elevators, mauna-loa-atmospheric, puma32H, stock_fardamento02, treasury, weather_izmir, wind.

- In Table 10, we showcase the **performance of various models on 18 high-dimensional datasets**. The results display the mean accuracy of different models, including ModernNCA (MNCA), MLP, KNN, RealMLP, XGBoost (XGB), Random Forest (RForest), Logistic Regression (LogReg), and TabPFN v2 (PFN-v2), along with variants like TabPFN v2-pca and TabPFN v2*. This highlights the ability of these models to handle high-dimensional data with many features.

- We demonstrate the **performance of various models on 12 multi-class classification tasks** with more than 10 classes in Table 11. The table provides the mean accuracy of models like KNN, TabPFN-v2*, XGBoost (XGB), CatBoost (CatB), Random Forest (RForest), ModernNCA (MNCA), MLP, Logistic Regression (LogReg), and RealMLP, showcasing how they perform on multi-class tasks with a larger number of classes. Additionally, we compare **PFN-v2-ECOC**, a

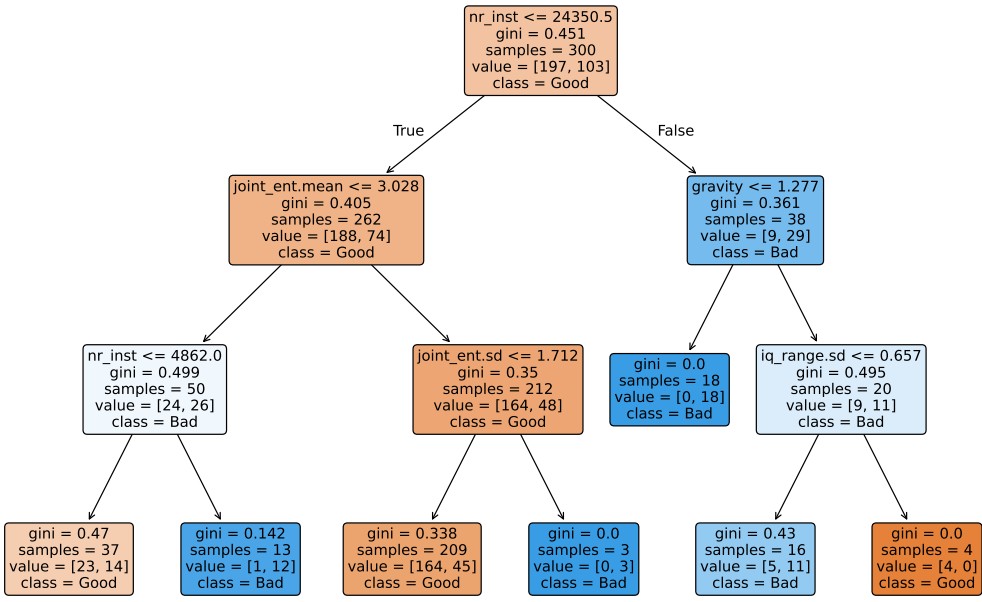

Figure 9: Decision tree for predicting TabPFN v2 performance (Good *vs.* Bad) based on dataset characteristics, constructed from experiments on 300 datasets. The tree splits on meta-features such as number of instances (`nr_inst`), joint entropy (`joint_ent.mean`), number of numerical features (`nr_num`), distance between minority and majority classes' center of mass (`gravity`) and interquartile range (`iq_range`) statistics. Each split is chosen to maximize information gain. The leaf nodes indicate the predicted performance class, the Gini impurity, and class distribution. This tree provides insights into the types of datasets where TabPFN v2 is expected to perform well.

multi-class classification solution provided by [28]. This method extends TabPFN-v2 by leveraging Error-Correcting Output Codes (ECOC) to enhance multi-class classification performance.[3]

- In Table 12, we compare the **performance of various models on 18 large-scale datasets**. The results show the mean accuracy or RMSE for MLP, Logistic/Linear Regression (LR), KNN, XGBoost (XGB), Random Forest (RForest), CatBoost (CatB), ModernNCA (MNCA), RealMLP, and different versions of TabPFN v2 (PFNv2, PFNv2 with K-means, PFNv2 with Bagging, and PFNv2*). This illustrates the models' performance on large-scale datasets.

- In Table 13, we show the **performance of TabPFN v2 and the extracted feature embeddings** across 29 classification datasets. The table includes average classification accuracy for each dataset when using feature embeddings from different transformer layers (Layer 6, Layer 9, Layer 12), as well as a combined approach where embeddings from multiple layers are concatenated. The "selected layers" column indicates the layers chosen based on validation set performance, offering insights into how different layers contribute to overall model performance. In addition to evaluating the performance of TabPFN v2 and the extracted feature embeddings, we also compared the results with embeddings obtained using the vanilla strategy (Vanilla).

Table 9: Main results of TabPFN v2 on 300 datasets, including accuracy (for classification tasks) and RMSE (for regression tasks), along with the corresponding mean and standard deviation for each dataset. Among the 300 datasets, 200 are classification datasets, and 100 are regression datasets. The results demonstrate the effectiveness of TabPFN v2 across both classification and regression tasks.

| Dataset | Mean + Std | Dataset | Mean + Std |
|---|---|---|---|
| ASP-POTASSCO-classification | 43.50 ± 1.27 | Amazon_employee_access | 94.22 ± 0.04 |
| BLE_RSSI_localization | 73.37 ± 0.15 | BNG(breast-w) | 98.56 ± 0.07 |

---

[3]https://github.com/PriorLabs/tabpfn-community/blob/main/src/tabpfn_extensions/many_class/many_class_classifier.py

| Dataset | Value | Dataset | Value |
|---|---|---|---|
| BNG(cmc) | 57.69 ± 0.17 | BNG(tic-tac-toe) | 79.42 ± 0.26 |
| Bank_Customer_Churn_Dataset | 87.53 ± 0.12 | Basketball_c | 70.65 ± 0.47 |
| California-Housing-Classification | 91.47 ± 0.17 | Cardiovascular-Disease-dataset | 72.92 ± 0.13 |
| Click_prediction_small | 83.29 ± 0.03 | Contaminant-10.0GHz | 94.42 ± 0.36 |
| Contaminant-10.5GHz | 95.17 ± 0.32 | Contaminant-11.0GHz | 93.93 ± 0.50 |
| Contaminant-9.0GHz | 93.01 ± 0.47 | Contaminant-9.5GHz | 93.21 ± 0.50 |
| Credit_c | 69.98 ± 0.15 | Customer_Personality_Analysis | 90.03 ± 0.21 |
| Diabetic_Retinopathy_Debrecen | 72.81 ± 1.07 | E-CommereShippingData | 67.54 ± 0.21 |
| Employee | 84.80 ± 0.30 | FICO-HELOC-cleaned | 75.35 ± 0.21 |
| FOREX_audcad-day-High | 74.51 ± 0.51 | FOREX_audcad-hour-High | 71.01 ± 0.20 |
| FOREX_audchf-day-High | 76.66 ± 0.45 | FOREX_audjpy-day-High | 78.00 ± 0.28 |
| FOREX_audjpy-hour-High | 71.41 ± 0.32 | FOREX_audsgd-hour-High | 69.81 ± 0.39 |
| FOREX_audusd-hour-High | 69.57 ± 0.48 | FOREX_cadjpy-day-High | 71.68 ± 0.53 |
| FOREX_cadjpy-hour-High | 70.55 ± 0.40 | Firm-Teacher-Direction | 84.42 ± 0.47 |
| Fitness_Club_c | 79.67 ± 0.24 | GAMETES_Epistasis | 68.75 ± 0.82 |
| GAMETES_Heterogeneity | 65.90 ± 1.84 | Gender_Gap_in_Spanish_WP | 60.58 ± 0.24 |
| GesturePhaseSegmentationProcessed | 71.36 ± 1.15 | HR_Analytics | 80.02 ± 0.13 |
| Heart-Disease-Dataset | 91.23 ± 0.54 | INNHotelsGroup | 87.98 ± 0.23 |
| Indian_pines | 96.41 ± 0.23 | Insurance | 75.75 ± 0.00 |
| Intersectional-Bias-Assessment | 94.73 ± 0.13 | Is-this-a-good-customer | 88.41 ± 0.00 |
| JapaneseVowels | 99.68 ± 0.08 | KDD | 80.14 ± 0.46 |
| KDDCup09_upselling | 81.06 ± 0.26 | Long | 99.88 ± 0.00 |
| MIC | 90.20 ± 0.56 | MagicTelescope | 88.13 ± 0.21 |
| Marketing_Campaign | 88.11 ± 0.41 | Mobile_Price_Classification | 97.10 ± 0.29 |
| Nutrition_Health_Survey | 83.45 ± 0.22 | Performance-Prediction | 73.23 ± 0.61 |
| PhishingWebsites | 96.74 ± 0.13 | PieChart3 | 87.31 ± 0.28 |
| Pima_Indians_Diabetes_Database | 75.93 ± 0.66 | PizzaCutter3 | 88.20 ± 0.45 |
| Pumpkin_Seeds | 87.93 ± 0.21 | QSAR_biodegradation | 88.50 ± 0.50 |
| Rain_in_Australia | 83.88 ± 0.11 | SDSS17 | 97.33 ± 0.06 |
| Shipping | 68.73 ± 0.40 | Telecom_Churn_Dataset | 95.18 ± 0.50 |
| UJI_Pen_Characters | 45.71 ± 2.16 | VulNoneVul | 98.95 ± 0.00 |
| Water_Quality_and_Potability | 65.49 ± 0.50 | Waterstress | 71.37 ± 0.96 |
| Wilt | 99.28 ± 0.06 | abalone | 63.58 ± 0.38 |
| accelerometer | 73.96 ± 1.32 | ada | 85.40 ± 0.25 |
| ada_agnostic | 83.99 ± 0.34 | ada_prior | 85.32 ± 0.19 |
| adult | 85.93 ± 0.12 | airlines_2000 | 62.28 ± 0.48 |
| allbp | 97.85 ± 0.19 | allrep | 98.65 ± 0.12 |
| analcatdata_authorship | 99.72 ± 0.29 | artificial-characters | 73.90 ± 0.99 |
| autoUniv-au4-2500 | 69.81 ± 1.00 | autoUniv-au7-1100 | 41.18 ± 1.58 |
| bank | 90.86 ± 0.19 | banknote_authentication | 55.64 ± 0.18 |
| baseball | 93.81 ± 0.40 | car-evaluation | 98.29 ± 0.22 |
| churn | 96.33 ± 0.28 | cmc | 59.59 ± 0.49 |
| company_bankruptcy_prediction | 97.33 ± 0.07 | compass | 71.05 ± 0.29 |
| connect-4 | 76.78 ± 0.35 | contraceptive_method_choice | 62.10 ± 0.37 |
| credit | 78.10 ± 0.11 | credit-g | 79.50 ± 0.81 |
| customer_satisfaction_in_airline | 94.79 ± 0.11 | dabetes_130-us_hospitals | 63.08 ± 0.07 |
| default_of_credit_card_clients | 82.63 ± 0.08 | delta_ailerons | 95.47 ± 0.09 |
| dis | 99.07 ± 0.14 | dna | 97.25 ± 0.20 |
| drug_consumption | 40.32 ± 0.00 | dry_bean_dataset | 92.76 ± 0.10 |
| eeg-eye-state | 98.34 ± 0.12 | electricity | 86.57 ± 0.45 |
| estimation_of_obesity_levels | 98.66 ± 0.24 | eucalyptus | 72.88 ± 1.17 |
| eye_movements | 77.03 ± 1.68 | eye_movements_bin | 67.28 ± 2.60 |
| first-order-theorem-proving | 61.12 ± 0.70 | gas-drift | 99.47 ± 0.04 |
| golf_play_dataset_extended | 92.60 ± 0.44 | helena | 33.32 ± 0.21 |
| heloc | 72.75 ± 0.20 | hill-valley | 98.33 ± 0.52 |
| house_16H | 88.55 ± 0.18 | htru | 97.95 ± 0.06 |
| ibm-employee-performance | 100.0 ± 0.00 | in_vehicle_coupon | 73.20 ± 0.35 |
| internet_firewall | 92.85 ± 0.30 | internet_usage | 54.34 ± 2.64 |
| jasmine | 81.34 ± 0.42 | jm1 | 81.32 ± 0.10 |
| jungle_chess_2pcs_raw_endgame | 85.97 ± 1.82 | kc1 | 86.65 ± 0.34 |
| kdd_ipums_la_97-small | 88.50 ± 0.12 | kr-vs-k | 78.46 ± 1.01 |
| kr-vs-kp | 99.64 ± 0.15 | kropt | 77.96 ± 0.63 |
| law-school-admission-bianry | 100.0 ± 0.00 | led24 | 73.29 ± 0.62 |
| led7 | 73.99 ± 0.31 | letter | 97.57 ± 0.10 |
| madeline | 90.72 ± 0.48 | mammography | 98.71 ± 0.05 |
| maternal_health_risk | 83.28 ± 0.64 | mfeat-factors | 96.98 ± 0.28 |
| mfeat-fourier | 89.85 ± 0.86 | mfeat-karhunen | 96.42 ± 0.24 |
| mfeat-morphological | 76.63 ± 0.50 | mfeat-pixel | 96.10 ± 0.32 |
| mfeat-zernike | 84.10 ± 0.87 | mice_protein_expression | 100.0 ± 0.00 |
| microaggregation2 | 62.80 ± 0.14 | mobile_c36_oversampling | 98.11 ± 0.08 |
| mozilla4 | 93.58 ± 0.16 | naticusdroid+android+permissions | 96.41 ± 0.10 |
| national-longitudinal-survey-binary | 100.0 ± 0.00 | okcupid_stem | 74.47 ± 0.12 |
| one-hundred-plants-margin | 88.56 ± 0.74 | one-hundred-plants-shape | 79.52 ± 0.72 |
| one-hundred-plants-texture | 90.94 ± 0.75 | online_shoppers | 90.65 ± 0.10 |
| optdigits | 98.59 ± 0.12 | ozone-level-8hr | 94.92 ± 0.25 |
| ozone_level | 97.86 ± 0.11 | page-blocks | 97.67 ± 0.10 |
| pc1 | 93.51 ± 0.46 | pc3 | 88.78 ± 0.27 |
| pc4 | 90.87 ± 0.36 | pendigits | 99.56 ± 0.06 |
| philippine | 84.20 ± 1.24 | phoneme | 88.47 ± 0.35 |
| pol | 98.80 ± 0.09 | predict_students_dropout | 78.11 ± 0.38 |
| rice_cammeo_and_osmancik | 92.74 ± 0.23 | ringnorm | 98.00 ± 0.13 |

| | | | |
|---|---|---|---|
| rl | 86.04 ± 0.44 | satimage | 92.30 ± 0.29 |
| segment | 93.91 ± 0.19 | seismic+bumps | 93.40 ± 0.08 |
| semeion | 92.41 ± 0.94 | shill-bidding | 90.31 ± 0.18 |
| shrutime | 86.97 ± 0.11 | shuttle | 99.86 ± 0.04 |
| spambase | 94.85 ± 0.19 | splice | 96.61 ± 0.22 |
| sports_articles | 84.93 ± 0.40 | statlog | 72.13 ± 0.97 |
| steel_plates_faults | 84.68 ± 0.55 | svmguide3 | 85.54 ± 0.54 |
| sylvine | 97.30 ± 0.27 | taiwanese_bankruptcy_prediction | 97.20 ± 0.07 |
| telco-customer-churn | 80.29 ± 0.28 | texture | 100.0 ± 0.00 |
| W thyroid | 99.48 ± 0.06 | thyroid-ann | 99.34 ± 0.08 |
| thyroid-dis | 68.75 ± 0.34 | turiye_student_evaluation | 51.74 ± 0.18 |
| twonorm | 97.94 ± 0.08 | vehicle | 84.31 ± 1.29 |
| walking-activity | 61.22 ± 0.22 | wall-robot-navigation | 99.44 ± 0.10 |
| water_quality | 90.12 ± 0.12 | waveform-5000 | 86.29 ± 0.26 |
| waveform_v1 | 86.59 ± 0.25 | website_phishing | 90.48 ± 0.48 |
| wine | 75.12 ± 0.72 | wine-quality-red | 58.35 ± 0.76 |
| wine-quality-white | 64.15 ± 0.69 | yeast | 60.18 ± 0.65 |
| 1000-Cameras-Dataset | 607.71 ± 6.61 | 2dplanes | 1.01 ± 0.00 |
| RSSI_Estimation | 0.00068 ± 0.00 | RSSI_Estimation1 | 0.00092 ± 0.00 |
| Abalone_reg | 2.08 ± 0.00 | Ailerons | 0.00015 ± 0.00 |
| Fiat | 716.20 ± 4.05 | BNG(echoMonths) | 11.41 ± 0.03 |
| BNG(lowbwt) | 455.27 ± 0.78 | BNG(mv) | 4.63 ± 0.01 |
| BNG(stock) | 2.95 ± 0.02 | Bias_correction_r | 0.60 ± 0.01 |
| Bias_correction_r_2 | 0.52 ± 0.01 | Brazilian_houses_reproduced | 0.01 ± 0.00 |
| CPMP-2015-regression | 478.02 ± 5.40 | CPS1988 | 364.02 ± 0.24 |
| CookbookReviews | 1.52 ± 0.02 | Data_Science_Salaries | 60237.28 ± 102.97 |
| Diamonds | 533.30 ± 6.30 | Facebook_Comment_Volume | 23.16 ± 0.20 |
| Food_Delivery_Time | 7.55 ± 0.03 | Goodreads-Computer-Books | 0.43 ± 0.00 |
| IEEE80211aa-GATS | 0.02 ± 0.00 | Job_Profitability | 13.14 ± 0.02 |
| bike_sharing_demand | 68.41 ± 0.60 | Laptop_Prices_Dataset | 439.87 ± 3.10 |
| Wave_Energy_Perth_100 | 15507.90 ± 104.31 | Wave_Energy_Sydney_100 | 14737.67 ± 150.43 |
| Wave_Energy_Sydney_49 | 4567.97 ± 64.02 | MIP-2016-regression | 20966.10 ± 454.90 |
| MiamiHousing2016 | 83101.09 ± 507.30 | Mobile_Phone_Market | 714.87 ± 11.15 |
| Moneyball | 19.42 ± 0.08 | NASA_PHM2008 | 40.24 ± 0.06 |
| NHANES_age_prediction | 15.47 ± 0.04 | OnlineNewsPopularity | 8606.54 ± 7.04 |
| Parkinson_Sound_Record | 14.58 ± 0.09 | Parkinsons_Telemonitoring | 0.60 ± 0.04 |
| Physicochemical_r | 3.45 ± 0.04 | SAT11-HAND-runtime | 1232.03 ± 58.01 |
| Shop_Customer_Data | 28.56 ± 0.01 | Superconductivty | 10.17 ± 0.07 |
| Wine_Quality_red | 0.65 ± 0.00 | Wine_Quality_white | 0.68 ± 0.00 |
| airfoil_self_noise | 1.16 ± 0.02 | analcatdata_supreme | 0.09 ± 0.00 |
| archive2 | 342.64 ± 3.20 | archive_r56_Portuguese | 2.86 ± 0.02 |
| auction_verification | 1145.54 ± 146.94 | avocado_sales | 0.09 ± 0.00 |
| bank32nh | 0.08 ± 0.00 | bank8FM | 0.03 ± 0.00 |
| boston | 4.25 ± 0.19 | chscase_foot | 0.95 ± 0.00 |
| colleges | 0.14 ± 0.00 | combined_cycle_power_plant | 3.22 ± 0.05 |
| communities_and_crime | 0.13 ± 0.00 | concrete_compressive_strength | 4.63 ± 0.07 |
| cpu_act | 2.65 ± 0.03 | cpu_small | 3.06 ± 0.02 |
| dataset_sales | 4.04 ± 0.02 | debutanizer | 0.04 ± 0.00 |
| delta_elevators | 0.0014 ± 0.00 | elevators | 0.0019 ± 0.00 |
| fifa | 0.78 ± 0.00 | fried | 1.01 ± 0.00 |
| garments_worker_productivity | 0.13 ± 0.00 | gas_turbine_emission | 0.44 ± 0.00 |
| healthcare_insurance_expenses | 4716.87 ± 36.52 | house_16H_reg | 29631.75 ± 251.56 |
| house_8L | 28617.41 ± 202.41 | house_prices_nominal | 30676.02 ± 2455.48 |
| house_sales_reduced | 132655.03 ± 1847.33 | houses | 42559.98 ± 928.78 |
| housing_price_prediction | 1009361.62 ± 8758.05 | kin8nm | 0.08 ± 0.00 |
| mauna-loa-atmospheric-co2 | 0.39 ± 0.01 | mv | 0.02 ± 0.00 |
| pol_reg | 3.84 ± 0.10 | pole | 3.21 ± 0.14 |
| puma32H | 0.01 ± 0.00 | puma8NH | 3.24 ± 0.00 |
| qsar_aquatic_toxicity | 1.05 ± 0.01 | qsar_fish_toxicity | 0.86 ± 0.01 |
| satellite_image | 0.65 ± 0.00 | sensory | 0.77 ± 0.01 |
| socmob | 19.53 ± 0.64 | space_ga | 0.09 ± 0.00 |
| steel_industry_energy | 0.37 ± 0.03 | stock | 0.65 ± 0.01 |
| stock_fardamento02 | 17.57 ± 0.08 | sulfur | 0.03 ± 0.00 |
| topo_2_1 | 0.03 ± 0.00 | treasury | 0.23 ± 0.00 |
| us_crime | 0.14 ± 0.00 | volume | 52.09 ± 0.34 |
| weather_izmir | 1.09 ± 0.01 | wind | 2.83 ± 0.00 |
| wine+quality | 0.72 ± 0.00 | yprop_4_1 | 0.03 ± 0.00 |

# F   Limitations

While this paper does not introduce a new model architecture or training paradigm, it offers a timely and principled analysis of TabPFN v2, a powerful tabular foundation model. Our contributions lie in empirically evaluating its strengths, identifying its limitations, and proposing practical extensions that enhance its applicability—particularly to large-scale, high-dimensional, and multi-class settings. A key limitation of our work is that the proposed extensions are primarily post-hoc and do not fully

Table 8: Meta-features used in the meta-feature analysis of TabPFN v2 performance.

| Meta-Feature | Explanation |
|---|---|
| attr_conc | The concentration coef. of each pair of distinct attributes. |
| class_conc | The concentration coefficient between each attribute and class. |
| class_ent | The target attribute Shannon's entropy. |
| inst_to_attr | The ratio between the number of instances and attributes. |
| mean | The mean value of each attribute. |
| sd | The standard deviation of each attribute. |
| var | The variance of each attribute. |
| range | The range (max - min) of each attribute. |
| iq_range | The interquartile range (IQR) of each attribute. |
| nr_attr | The total number of attributes. |
| sparsity | The (possibly normalized) sparsity metric for each attribute. |
| t_mean | The trimmed mean of each attribute. |
| nr_bin | The number of binary attributes. |
| nr_cat | The number of categorical attributes. |
| nr_num | The number of numeric features. |
| nr_norm | The number of attributes normally distributed based in a given method. |
| nr_cor_attr | The number of distinct highly correlated pair of attributes. |
| gravity | The distance between minority and majority classes' center of mass. |
| nr_class | The number of distinct classes. |
| joint_ent | The joint entropy between each attribute and class. |
| attr_ent | Shannon's entropy for each predictive attribute. |
| cov | The absolute value of the covariance of distinct dataset attribute pairs. |
| eigenvalues | The eigenvalues of covariance matrix from dataset. |
| eq_num_attr | The number of attributes equivalent for a predictive task. |
| max | The maximum value from each attribute. |
| min | The minimum value from each attribute. |
| median | The median value from each attribute. |
| freq_class | The relative frequency of each distinct class. |
| mad | The Median Absolute Deviation (MAD) adjusted by a factor. |
| mad | The Median Absolute Deviation (MAD) adjusted by a factor. |
| mut_inf | The mutual information between each attribute and target. |
| nr_inst | The number of instances (rows) in the dataset. |
| nr_outliers | The number of attributes with at least one outlier value. |
| ns_ratio | The noisiness of attributes. |
| imblance_ratio | The ratio of the number of instances in the minority to the majority class. |
| attr_to_inst | The ratio between the number of attributes. |

address the scalability constraints inherent to the original architecture. Nevertheless, by improving model usability without retraining, we reduce the computational and environmental costs typically associated with large model development.

To the best of our knowledge, this work poses no explicit ethical concerns. It provides practical guidance for applying pre-trained tabular models in real-world domains such as healthcare and finance, where transparency and efficiency are essential. Our study highlights the value of understanding and extending foundation models alongside architectural innovation.

Table 10: Performance of various models on 18 high-dimensional datasets. The results show the mean accuracy of different models, including ModernNCA (MNCA), MLP, KNN, RealMLP, XGBoost (XGB), Random Forest (RForest), Logistic Regression (LogReg), TabPFN v2 (PFN-v2), TabPFN v2 with PCA (v2-pca), TabPFN v2 with subsampling (v2*), ProtoGate (ProtoG), and CatBoost (CatB). The performance is evaluated on high-dimensional datasets, with the values representing mean accuracy for each model.

| Dataset | MNCA | MLP | KNN | RealMLP | XGB | RForest | LogReg | PFN-v2 | v2-pca | v2* | ProtoG | CatB |
|---|---|---|---|---|---|---|---|---|---|---|---|---|
| CLL_SUB_111 | 62.90 | 72.46 | 57.39 | 70.43 | 73.04 | 70.14 | 73.91 | 70.14 | 57.68 | 71.59 | 65.51 | 71.59 |
| BASEHOCK | 96.31 | 97.01 | 71.88 | 97.46 | 95.29 | 96.73 | 96.99 | 69.09 | 97.41 | 97.36 | 96.32 | 95.87 |
| Prostate_GE | 81.27 | 86.98 | 80.00 | 87.94 | 89.52 | 87.94 | 91.43 | 95.24 | 88.57 | 94.29 | 84.13 | 94.92 |
| PCMAC | 88.21 | 88.53 | 66.48 | 90.15 | 91.64 | 92.20 | 87.15 | 92.70 | 90.76 | 90.14 | 88.21 | 92.01 |
| GLI_85 | 81.57 | 85.49 | 76.47 | 89.80 | 82.35 | 83.92 | 90.59 | 80.39 | 86.27 | 92.55 | 81.96 | 80.78 |
| RELATHE | 88.18 | 90.54 | 75.03 | 90.23 | 87.11 | 87.30 | 90.49 | 86.36 | 87.65 | 89.95 | 89.92 | 90.35 |
| SMK_CAN_187 | 63.51 | 66.84 | 69.47 | 69.82 | 66.49 | 70.70 | 72.11 | 71.05 | 71.75 | 72.10 | 70.71 | 71.40 |
| warpPIE10P | 98.41 | 99.05 | 92.38 | 100.0 | 94.92 | 98.57 | 100.0 | 100.0 | 100.0 | 100.0 | 97.79 | 98.89 |
| leukemia | 90.22 | 95.11 | 86.67 | 94.67 | 97.78 | 92.00 | 96.00 | 92.44 | 93.33 | 96.00 | 94.00 | 94.22 |
| orlraws10P | 97.67 | 98.33 | 92.00 | 99.00 | 84.33 | 99.00 | 99.00 | 92.00 | 99.33 | 99.67 | 92.67 | 99.00 |
| GLIOMA | 58.00 | 60.67 | 68.00 | 67.33 | 66.67 | 64.00 | 64.00 | 62.67 | 69.33 | 68.67 | 69.91 | 66.67 |
| warpAR10P | 83.08 | 85.64 | 53.08 | 97.44 | 81.28 | 87.18 | 97.69 | 90.77 | 95.38 | 96.67 | 90.04 | 87.44 |
| TOX_171 | 76.00 | 88.19 | 70.86 | 90.48 | 78.10 | 78.67 | 90.29 | 80.95 | 82.48 | 87.24 | 85.52 | 83.05 |
| lung | 91.54 | 95.45 | 93.66 | 95.28 | 93.66 | 92.68 | 95.12 | 95.28 | 93.50 | 95.61 | 95.43 | 93.01 |
| ALLAML | 87.56 | 95.56 | 81.33 | 96.89 | 96.00 | 96.44 | 92.00 | 92.89 | 93.78 | 94.67 | 91.14 | 94.67 |
| colon | 78.46 | 78.97 | 76.92 | 83.08 | 74.87 | 82.56 | 86.15 | 81.54 | 78.46 | 79.49 | 78.46 | 77.95 |
| gisette | 97.21 | 97.57 | 95.04 | 97.86 | 97.55 | 96.82 | 97.51 | 97.35 | 97.26 | 97.23 | 97.18 | 97.78 |
| arcene | 81.67 | 85.50 | 84.50 | 81.00 | 75.00 | 86.83 | 88.00 | 83.67 | 88.33 | 92.00 | 85.33 | 85.00 |
| Mean | 82.86 | 87.11 | 77.29 | 88.83 | 84.76 | 86.87 | 89.36 | 85.25 | 87.29 | 89.73 | 86.37 | 87.48 |

Table 11: Performance of various models on 12 multi-class classification tasks with more than 10 classes. The results show the mean accuracy of different models, including KNN, PFN-v2*, PFN-v2-ECOC, XGB (XGBoost), CatBoost (CatB), Random Forest (RForest), ModernNCA (MNCA), Multi-layer Perceptron (MLP), Logistic Regression (LogReg), and RealMLP. The performance is evaluated on 12 multi-class datasets with more than 10 classes, with accuracy values presented for each model on the respective datasets.

| Dataset | KNN | PFN-v2* | PFN-v2-DPT | PFN-v2-ECOC | XGB | CatB | RForest | MNCA | MLP | LogReg | RealMLP |
|---|---|---|---|---|---|---|---|---|---|---|---|
| 100-plants-texture | 79.69 | 90.94 | 82.67 | 84.92 | 77.06 | 89.73 | 82.65 | 80.52 | 83.92 | 86.88 | 88.35 |
| 100-plants-margin | 77.50 | 88.56 | 81.94 | 79.40 | 74.25 | 84.06 | 82.79 | 77.60 | 80.44 | 79.69 | 83.58 |
| 100-plants-shape | 60.31 | 79.52 | 72.15 | 63.38 | 56.15 | 65.19 | 64.33 | 70.10 | 47.33 | 65.94 | 72.08 |
| UJI_Pen_Characters | 36.26 | 45.71 | 33.38 | 44.20 | 30.35 | 38.88 | 34.24 | 44.03 | 37.75 | 19.41 | 46.37 |
| texture | 98.45 | 100.0 | 99.98 | 100.0 | 98.55 | 99.13 | 96.76 | 99.68 | 99.40 | 99.64 | 99.95 |
| letter | 94.90 | 97.57 | 96.69 | 97.78 | 96.26 | 96.75 | 91.56 | 97.96 | 96.40 | 75.80 | 98.31 |
| walking-activity | 60.29 | 61.22 | 57.28 | 61.92 | 65.06 | 64.92 | 61.74 | 64.85 | 60.64 | 27.02 | 65.13 |
| helena | 28.94 | 33.31 | 28.54 | 19.20 | 32.42 | 37.90 | 33.91 | 36.58 | 37.91 | 33.40 | 38.55 |
| internet_usage | 30.17 | 54.34 | 50.51 | 50.86 | 51.08 | 37.90 | 33.91 | 52.09 | 43.00 | 37.73 | 52.23 |
| kropt | 71.22 | 77.96 | 71.44 | 77.11 | 86.95 | 79.26 | 71.77 | 78.27 | 64.45 | 28.08 | 92.03 |
| kr-vs-k | 70.78 | 78.46 | 71.54 | 76.29 | 87.26 | 74.81 | 71.60 | 76.83 | 65.03 | 28.03 | 91.85 |
| ASP-POTASSCO | 34.75 | 43.50 | 41.88 | 45.27 | 42.24 | 41.08 | 42.86 | 37.45 | 29.63 | 35.14 | 41.70 |
| Mean | 61.94 | 70.93 | 65.67 | 66.69 | 66.47 | 67.47 | 64.01 | 68.00 | 62.16 | 51.40 | 72.51 |

Table 12: Performance of various models on 18 large-scale datasets. The results show the mean accuracy/RMSE of different models, including MLP, Logistic Regression/Linear Regression (LR), KNN, XGBoost (XGB), Random Forest (RForest), CatBoost (CatB), ModernNCA (MNCA), RealMLP, and various versions of TabPFN v2: original TabPFN v2 (PFNv2), TabPFN v2 with K-means (PFNv2-K), TabPFN v2 with Bagging (PFNv2-B), PFNv2* (TabPFNv2*), PFNv2-DT (TabPFN-DT), and PFNv2-DF (TabPFN-DF).

| Dataset | MLP | LR | KNN | XGB | RForest | CatB | MNCA | RealMLP | PFNv2 | PFNv2-K | PFNv2-B | PFNv2* | PFNv2-DT | PFNv2-DF |
|---|---|---|---|---|---|---|---|---|---|---|---|---|---|---|
| BNG(credit-a) | 90.07 | 85.98 | 87.41 | 90.21 | 89.25 | 91.13 | 89.98 | 90.91 | 89.55 | 89.01 | 89.66 | 89.89 | 90.45 | 90.43 |
| CDC_Indicators | 86.79 | 86.55 | 86.39 | 86.76 | 86.60 | 86.78 | 86.76 | 86.76 | 86.65 | 86.68 | 86.69 | 86.74 | 86.75 | 86.70 |
| Higgs | 75.53 | 64.29 | 65.16 | 73.33 | 71.87 | 74.81 | 73.28 | 75.36 | 71.64 | 71.56 | 72.01 | 72.13 | 73.53 | 73.62 |
| Smoking_signal | 73.90 | 72.53 | 72.36 | 73.87 | 73.08 | 73.99 | 73.63 | 74.00 | 73.47 | 73.37 | 73.55 | 73.69 | 73.74 | 73.84 |
| nomao | 96.19 | 94.59 | 95.20 | 96.92 | 96.07 | 97.03 | 96.68 | 96.37 | 96.08 | 96.29 | 96.12 | 96.18 | 96.75 | 96.34 |
| sf-police-incidents | 87.84 | 87.84 | 85.87 | 87.68 | 87.84 | 87.87 | - | 87.84 | 87.84 | 87.84 | 87.84 | 87.84 | 87.84 | 87.84 |
| Data_Crowdfunding | 96.48 | 67.04 | 93.70 | 96.89 | 95.29 | 96.81 | 96.53 | 96.71 | 94.59 | 91.81 | 94.96 | 95.07 | 96.90 | 96.83 |
| Fashion-MNIST | 89.54 | 85.69 | 86.00 | 90.03 | 86.57 | 90.24 | 89.36 | 90.25 | 68.40 | 82.82 | 83.89 | 86.26 | 78.91 | 78.82 |
| covertype | 94.01 | 72.54 | 92.76 | 96.30 | 78.30 | 90.77 | 97.31 | 97.38 | 83.54 | 82.95 | 84.16 | 86.85 | 97.38 | 97.44 |
| jannis | 71.99 | 64.60 | 65.67 | 71.83 | 69.19 | 72.26 | 72.57 | 73.00 | 70.24 | 70.26 | 70.59 | 71.31 | 72.57 | 72.50 |
| poker-hand | 99.99 | 50.12 | 54.01 | 99.51 | 64.63 | 97.69 | 76.31 | 99.88 | 41.97 | 38.86 | 36.80 | 54.12 | 91.13 | 92.33 |
| volkert | 69.85 | 58.75 | 67.41 | 69.74 | 62.71 | 70.88 | 77.18 | 73.76 | 62.82 | 62.15 | 62.81 | 64.84 | 68.66 | 67.76 |
| Airlines_DepDelay ($\times 10^1$) | 2.905 | 2.933 | 3.170 | 2.891 | 2.907 | 2.881 | - | 2.482 | 2.937 | 2.933 | 2.937 | 2.915 | 2.900 | 2.897 |
| Wave_Energy_Farm ($\times 10^3$) | 8.199 | 13.19 | 32.29 | 6.917 | 7.294 | 7.173 | 6.148 | 59.05 | 7.214 | 8.375 | 7.063 | 10.506 | 6.616 | 6.785 |
| UJIndoorLoc ($\times 10^0$) | 9.958 | $\infty$ | 9.004 | 10.47 | 23.19 | 9.139 | 5.990 | 65.34 | 66.49 | 7.825 | 7.435 | 9.538 | 14.404 | 7.472 |
| blogfeedback ($\times 10^1$) | 2.387 | $\infty$ | 2.410 | 2.093 | 2.026 | 2.044 | 1.953 | 2.105 | 3.073 | 2.687 | 2.700 | 2.014 | 1.914 | 1.944 |
| microsoft ($\times 10^{-1}$) | 7.577 | 7.782 | 8.284 | 7.514 | 7.566 | 7.453 | 7.573 | 5.077 | 7.735 | 7.981 | 7.720 | 7.612 | 7.944 | 7.728 |
| yahoo ($\times 10^{-1}$) | 7.692 | 7.997 | 8.504 | 7.629 | - | 7.514 | - | 5.671 | 8.148 | 8.332 | 8.132 | 7.961 | 16.409 | 8.069 |

Table 13: Performance of TabPFN v2 and the extracted feature embeddings across 29 classification datasets. The table shows the average classification accuracy for each dataset when using different layers (Layer 6, Layer 9, Layer 12) of the transformer as feature embeddings, as well as the "combined" approach, where embeddings from up to three selected layers are concatenated. The "selected layers" column indicates the specific layers chosen for each dataset based on validation set performance. "Vanilla" refers to the embeddings extracted using the vanilla strategy, which utilizes only the 12th layer of the transformer. "S" and "P" refer to *unsupervised* embedding extraction approaches by appending a column of dummy labels with zero values and permuting each column as labels, respectively, as described in Appendix C.

| | PFN-v2 | Vanilla | S | P | layer-6 | layer-9 | layer-12 | combined | selected layers |
|---|---|---|---|---|---|---|---|---|---|
| FOREX_audchf-day-High | 77.38 | 50.68 | 56.95 | 69.48 | 68.39 | 73.57 | 74.11 | 77.11 | (5, 9, 11) |
| taiwanese_bankruptcy_prediction | 96.99 | 56.45 | 96.77 | 95.75 | 97.14 | 96.77 | 97.07 | 97.14 | (6) |
| rl | 85.51 | 50.00 | 60.56 | 70.82 | 66.90 | 69.52 | 86.72 | 87.53 | (11, 12) |
| pc3 | 89.46 | 10.22 | 89.78 | 86.90 | 90.10 | 88.82 | 88.82 | 88.82 | (8) |
| eye_movements_bin | 61.83 | 50.00 | 55.12 | 57.95 | 59.72 | 59.40 | 62.16 | 62.16 | (6, 9, 12) |
| BNG(breast-w) | 98.43 | 69.51 | 97.60 | 98.51 | 98.34 | 98.46 | 98.67 | 98.51 | (6, 9) |
| FOREX_cadjpy-hour-High | 69.53 | 51.79 | 66.55 | 71.12 | 62.12 | 64.87 | 70.66 | 70.88 | (4, 5, 6) |
| dis | 99.34 | 85.43 | 98.41 | 98.54 | 98.41 | 98.28 | 99.34 | 99.47 | (4, 5, 6) |
| sylvine | 97.46 | 85.66 | 72.78 | 95.71 | 92.49 | 93.95 | 97.27 | 96.49 | (1, 11) |
| BNG(tic-tac-toe) | 78.04 | 34.71 | 71.41 | 73.79 | 73.96 | 73.71 | 78.75 | 79.03 | (5, 10, 12) |
| online_shoppers | 90.59 | 84.51 | 85.93 | 89.46 | 90.02 | 90.11 | 90.63 | 90.02 | (8) |
| Cardiovascular-Disease-dataset | 72.84 | 50.86 | 68.73 | 72.60 | 72.96 | 73.06 | 73.14 | 73.09 | (5, 8, 12) |
| credit | 78.04 | 62.31 | 75.86 | 78.31 | 77.62 | 77.80 | 77.95 | 77.59 | (4, 6, 9) |
| FOREX_audsgd-hour-High | 67.26 | 51.48 | 65.49 | 70.14 | 57.24 | 61.06 | 69.62 | 70.41 | (7, 10, 12) |
| waveform-5000 | 86.00 | 80.60 | 55.70 | 87.10 | 85.60 | 85.60 | 86.40 | 86.90 | (1, 6, 11) |
| jungle_chess | 85.65 | 39.60 | 64.12 | 72.14 | 78.55 | 80.44 | 86.66 | 86.85 | (10, 11, 12) |
| BNG(cmc) | 57.40 | 42.62 | 52.48 | 55.16 | 56.19 | 56.72 | 57.72 | 57.88 | (9, 10, 12) |
| page-blocks | 97.35 | 94.25 | 95.43 | 96.35 | 96.07 | 96.71 | 97.17 | 97.35 | (6, 7, 12) |
| segment | 93.07 | 72.29 | 69.26 | 87.23 | 91.99 | 88.10 | 93.51 | 92.64 | (1, 12) |
| website_phishing | 90.77 | 36.90 | 82.66 | 90.04 | 85.98 | 87.08 | 91.88 | 91.88 | (7, 10) |
| baseball | 93.66 | 78.73 | 92.54 | 92.16 | 93.28 | 94.03 | 93.66 | 95.15 | (10, 11) |
| pendigits | 99.50 | 59.75 | 72.40 | 98.18 | 92.81 | 93.04 | 99.41 | 99.45 | (3, 4, 12) |
| Gender_Gap_in_Spanish_WP | 60.84 | 33.68 | 59.47 | 60.84 | 59.68 | 60.32 | 60.53 | 60.84 | (2, 12) |
| wine-quality-white | 62.35 | 10.51 | 49.29 | 55.10 | 54.08 | 55.31 | 63.57 | 64.39 | (8, 11, 12) |
| satimage | 91.21 | 82.04 | 84.99 | 89.19 | 88.72 | 88.65 | 91.91 | 91.91 | (8, 11, 12) |
| mfeat-fourier | 90.00 | 55.50 | 46.75 | 85.75 | 77.75 | 82.25 | 89.50 | 89.50 | (2, 7, 12) |
| VulNoneVul | 98.95 | 1.05 | 98.95 | 98.33 | 98.95 | 98.95 | 98.95 | 98.95 | (1) |
| law-school-admission-bianry | 100.0 | 99.83 | 79.76 | 98.82 | 100.0 | 100.0 | 100.0 | 100.0 | (6) |
| KDD | 80.34 | 78.45 | 62.36 | 76.76 | 79.34 | 78.35 | 81.23 | 79.94 | (1, 8, 10) |

