# OpenReview forum: "A Closer Look at TabPFN v2: Understanding Its Strengths and Extending Its Capabilities"
_NeurIPS.cc/2025/Conference — NeurIPS 2025 poster_

### Official Review · Reviewer_xoSe · 2025-06-06

**Clarity:** 4
**Significance:** 3
**Originality:** 2
**Rating:** 5
**Confidence:** 3

**Summary:**

This paper presents a detailed empirical analysis of the recently proposed TabPFN v2. The authors investigate the model's mechanisms for handling data heterogeneity and its performance on standard benchmarks. They demonstrate that TabPFN v2 can effectively infer attribute relationships even with randomized token inputs and can be repurposed as a powerful feature extractor that maps data into a linearly separable space. Furthermore, the paper identifies the model's known limitations concerning high-dimensional, large-scale, and many-category datasets. To mitigate these issues, the authors propose and evaluate several post-hoc, "divide-and-conquer" strategies, such as feature subsampling, instance subsampling, and task decomposition, showing that these methods can improve TabPFN v2's performance in these challenging regimes without model retraining.

**Questions:**

See Weakness

**Ethical Concerns:**

["NO or VERY MINOR ethics concerns only"]

**Final Justification:**

After reading the author's response and the comments of other reviewers, I am inclined to maintain the rating, as I have already given it a positive rating.

**Limitations:**

Yes

**Quality:**

3

**Strengths And Weaknesses:**

**Strengths:**

1. The paper's primary strength lies in its extensive and rigorous empirical evaluation. The authors conduct experiments on a very large suite of datasets (273 for the main analysis) and compare against a wide range of baselines. The analysis of the model's internal workings, through visualizations of embedding spaces and attention maps, is detailed and provides valuable phenomenological insights.

2. The paper is exceptionally well-written, well-structured, and easy to follow. The research objectives are stated clearly in the introduction, and the narrative flows logically from analyzing TabPFN v2's strengths to identifying its weaknesses and then proposing solutions. The figures and tables are informative and support the main claims effectively.

**Weaknesses:**

1. The core contribution is not the proposal of a new method, but the analysis of an existing model and the application of well-established, standard machine learning techniques to mitigate its known weaknesses. For example, The method for high-dimensional data is essentially feature subsampling and ensembling (bagging), a core idea behind Random Forests.

2. The work feels more like a technical report or an extended ablation study that would have been suitable for the evaluation track. The attempt to create a theoretical framing by linking the methods to "Chain-of-Thought" prompting is superficial and unsubstantiated, as the underlying mechanisms are entirely different.

---

> ### Author Rebuttal · Authors · 2025-07-31
>
> We sincerely thank Reviewer xoSe for the positive feedback highlighting our paper’s extensive empirical evaluation, in-depth model analysis, and clear presentation.
>
> We respectfully address the two concerns as follows:
>
> **Q1: On the novelty of our contributions**:
>
> While we do not introduce a new model architecture, our central contribution lies in systematically evaluating, understanding, and extending TabPFN v2—a model that is rapidly becoming foundational in the tabular domain. As argued in the main text, principled analysis and adaptation of such pre-trained models is increasingly impactful and necessary, especially as foundation models become more opaque and widely adopted. Our extensions (e.g., TabPFN v2*) are not just reapplications of known techniques like bagging, but carefully tailored adaptations that address the scalability limitations of TabPFN v2 without retraining, enabling its use in previously unreachable domains like high-dimensional or many-class tasks.
>
> **Q2: On the connection to Chain-of-Thought (CoT)**:
>
>  We acknowledge that the CoT analogy is conceptual rather than mechanistic. As clarified in Section 7, the idea is that sequential model reuse (e.g., using TabPFN v2 in a divide-and-conquer manner) mimics the layered reasoning process in CoT prompting. We agree this could be better framed and will revise the language to avoid overstating the parallel.
>
> We hope this clarifies the intention and value of our work as a comprehensive, principled effort to advance the use and understanding of tabular foundation models.

---

> > ### Comment · Reviewer_xoSe · 2025-08-01
> > **Follow up**
> >
> > Thank you for your response. After reading the author's response and the comments of other reviewers, I am inclined to maintain the rating, as I have already given it a positive rating.

---

> > > ### Author Response · Authors · 2025-08-07
> > > **Re: Follow up**
> > >
> > > Dear Reviewer xoSe,
> > >
> > > Thank you for your prompt feedback. We are glad to know that you are inclined to maintain your positive rating (originally, 5: Accept). We will incorporate our rebuttal into the final version, and we would appreciate it if you would be supportive of the acceptance of our paper.

---

### Official Review · Reviewer_vnsZ · 2025-07-02

**Clarity:** 4
**Significance:** 2
**Originality:** 3
**Rating:** 4
**Confidence:** 3

**Summary:**

The authors propose a systematic study of TabPFN v2, a popular ICL tabular foundation model.
More specifically, they analyze how this model learns feature dependencies, show that it can be repurposed into an encoder and propose some test-time strategies to improve performance.

**Questions:**

See question in the text above

**Ethical Concerns:**

["NO or VERY MINOR ethics concerns only"]

**Limitations:**

yes

**Quality:**

3

**Strengths And Weaknesses:**

As tabular ICL foundation model becomes more widespread understand how they work, can be used and improved becomes more important.
First, it is appreciated to have an overview of some of the internal mechanics of TabPFN v2 publicly available as the nature paper is quite scarce on details.

**Attribute relationships**: It is interesting to see that features tend to attend to target first then to other features. Did you have more insights or other methods of explanation on feature relationship? It can be hard to relate attention maps to explanations. For example does a pair of attributes with high attention weight really contributes more to model performance than one with lower one?

**TabPFN as a feature encoder**: While I agree that naive feature encoding will fail and that you method is sound, I am wondering, what happens if you simply use context (X_train, y_train) and queries = (X_train, X_test) and use embeddings of the queries for train and test, i.e no partitioning. Is that a bad feature encoder? Intuitively the network probably doesn't "replace" right away the dummy label of a train example in Q by its true label in S so I would have actually assumed this to perform decently.

Another question, this discrepancy disappears through the layers as the test examples get closer to training ones (fig 4 in layer layers), so does the naive approach actually also work decently for later layers?

**Improvements**: this is the most interesting section to me, I like that you did some good ablations comparing different methods.
Couple questions.
For 5.b, you are doing $\sqrt{C}$ ensemblings vs $\log_{10) C$: did you control for the number of ensembles in any way? Or does PFNv2* use more compute (and maybe even more ensembles over features and so on?).
For 5a I am not too sure this can be called divide and conquer: this usually implies solving the subproblems recursively, this seems to be a simple Monte Carlo ensembling no?
Is 5c also compute controlled? The decision forest/SQ seem quite more expensive than the rest.

Overall the novelty of the paper is not very high, but the content is clear, exposes some important ideas, test them extensively and has some nice insights.

---

> ### Author Rebuttal · Authors · 2025-07-31
>
> We sincerely thank Reviewer vnsZ for the thoughtful and constructive feedback. We appreciate your recognition of the value in analyzing and understanding tabular foundation models, particularly in clarifying the internal mechanisms of TabPFN v2. We address your specific questions and comments below:
>
> **Q1: Attribute relationships and attention maps**
>
> This is a very good question. In our humble opinion, understanding the inner workings of transformers—especially in the tabular context—remains an open and actively evolving research area, encompassing topics such as interpretability, explainability, and mechanistic understanding. In this work, we primarily rely on the most widely adopted approach: analyzing attention maps. While this provides some qualitative insights into how TabPFN v2 models feature interactions, we agree that attention alone may not fully capture the model’s reasoning process.
>
> We plan to explore more advanced interpretability techniques in future work. Potential directions include probing-based analyses, attribution methods (e.g., SHAP adapted for in-context learning), and layer-wise relevance propagation. However, these approaches are still under development for transformer-based tabular models, and challenges remain in defining reliable ground truth for feature importance and interpreting high-dimensional interactions in models trained across diverse datasets. We appreciate the reviewer’s encouragement to pursue these avenues.
>
> **Q2: Using the same context and query without partitioning**
>
> Indeed, we experimented with this variant where the context contains all training data, and the query set includes both training and test points. This approach improves upon the vanilla feature extraction baseline but still underperforms compared to our proposed leave-one-fold-out method. We refer to this as the “non-partitioned” strategy, which suffers from role ambiguity—query points that appear in the support set (with dummy or ground-truth labels) are treated inconsistently. The network may not fully differentiate between training and test roles, degrading feature consistency. The performance ranking (lower is better) on classification tasks is as follows:
>
> | Method                     | Avg. Rank |
> |----------------------------|-----------|
> | Leave-one-fold-out (Ours)                       | **2.10**     |
> | TabPFN v2            | 2.72      |
> | Non-partitioned (12 layers)| 3.07      |
> | Non-partitioned (9 layers) | 4.29      |
> | Non-partitioned (6 layers) | 4.43      |
> | Vanilla (12 layers)        | 5.86      |
> | Vanilla (9 layers)         | 6.62      |
> | Vanilla (6 layers)         | 6.90      |
>
> **Q3: Does the naive strategy improve in later layers?**
>
> Yes, embeddings from the vanilla extraction do become smoother and more clustered in deeper layers. However, their quality, as measured by downstream linear probing, still lags behind both TabPFN’s native outputs and our leave-one-fold-out embeddings. This suggests that deeper layers alone cannot fully mitigate the role ambiguity caused by naive embedding extraction.
>
> **Q4: Are the ensemble sizes in Figure 5b controlled?**
>
> Yes. For a fair comparison in many-class classification (Figure 5b), all methods, including TabPFN v2*, TabPFN v2-DPT, and our ensemble extensions, use the same number of ensemble members. For instance, TabPFN v2* uses √C ensembles, where C is the number of classes.
>
> **Q5: Is “divide-and-conquer” a fair term for 5a?**
>
> Thank you for pointing this out. While our method does not follow a recursive divide-and-conquer paradigm, we use the term more broadly to describe the process of decomposing complex problems (e.g., many-class classification) into simpler subproblems (e.g., binary digit classification), inspired by ideas in test-time strategies and Monte Carlo ensembling. We agree that clarification is needed and will revise the terminology in the final version.
>
> **Q6: Is compute controlled in 5c (TabPFN v2-DF)?**
>
> Yes, we carefully controlled compute budgets to ensure a fair comparison across all methods. Specifically, we set the number of ensembles to four for each method in the large-scale classification setting, including our PFN*, TabICL, PFN-Bagging, matching the cost of four forward passes of TabPFN v2. We will clarify this further in the final version of the paper.
>
> Below, we report a subset of average rank results across large-scale classification datasets, demonstrating that our proposed PFN\* remains competitive under controlled compute:
>
> | Method                | Avg. Rank |
> |-----------------------|-----------|
> | CatBoost              | 2.83      |
> | RealMLP               | 2.92      |
> | **PFN\*-DF** (ours)      | **4.50**  |
> | XGBoost               | 4.67      |
> | ModernNCA             | 4.73      |
> | TabICL             | 6.25      |
> | PFN\*-SQ              | 8.42      |
> | PFN-v2-Bagging      | 10.25     |
> | RandomForest          | 11.08     |
> | PFN-v2                | 11.42     |
>
> These results confirm that PFN\* achieves strong performance even with strict compute constraints. We sincerely thank the reviewer for raising this point and helping improve the clarity and rigor of our experimental design.
>
> In light of our further clarification, we would appreciate it if the reviewer would consider increasing the rating. We would be happy to address additional questions you may have in the author-reviewer discussion phase.

---

> > ### Comment · Reviewer_vnsZ · 2025-08-07
> > **Answer to rebuttal**
> >
> > Thank you for your precisions and additional details.
> >
> > I realize this is late in the discussion period so this won't affect my score but I have tried the subsampling vs PCA and class ensembling on top of the class to digit mapping. I have tried this with another model (think TabICL/TabDPT) on about 70+ datasets (CC18 suite) and have found the improvements to be quite small (or even negative often in the case of PCA vs subsampling). It may be something to do with my setup (fewer large class/high dim datasets), but I think having confidence intervals or doing a critical difference diagram might be beneficial to assess to statistical robustness of some of your conclusions.
> >
> > I am positively inclined towards this work, however I still have concerns about the overall novelty. I will elect to keep my score as is as it reflects my overall opinion.

---

> > > ### Author Response · Authors · 2025-08-07
> > > **Re: Answer to rebuttal by Reviewer vnsZ**
> > >
> > > We sincerely thank the reviewer for the additional feedback and for taking the time to experiment with our proposed strategies, even late in the discussion period.
> > >
> > > We note that the effectiveness of subsampling and class ensembling is more pronounced in high-dimensional (>2,000) and many-class (>10) settings, respectively — these are where the original TabPFN v2 underperforms, and the focus of our Section 7. We thus have two hypotheses regarding your findings. First, the 70+ datasets in the CC18 suite are generally low-dimensional and below 10 classes (except CIFAR-10 with high dimensionality, and five other datasets with more than 10 classes). Second, the application of subsampling and class ensembling to TabICL or TabDPT may require some modifications. We sincerely appreciate that you shared your findings with us, and we will further discuss this in the final version. We will also consider including confidence intervals.
> > >
> > > Once again, thank you for the constructive feedback and for acknowledging the practical contributions of our work despite your concerns about novelty. We greatly appreciate your positive inclination toward our work. Your insights have helped us better articulate the scope and implications of our findings.

---

> ### Author Response · Authors · 2025-08-06
>
> Dear Reviewer,
>
> Thank you for taking the time to review our submission. We sincerely appreciate your feedback and have done our best to address your comments and concerns in the author response.As the discussion period is ongoing, we would like to kindly check in to see if you had a chance to review our rebuttal and whether you have any further questions or require clarifications. We are happy to provide additional information or engage in discussion to help improve the quality and clarity of the paper.
>
> Thank you again for your time and effort in the review process.
>
> Best regards,
>
> Authors

---

### Official Review · Reviewer_pvYD · 2025-07-03

**Clarity:** 3
**Significance:** 3
**Originality:** 2
**Rating:** 4
**Confidence:** 4

**Summary:**

The paper presents an in-depth empirical and analytical study of TabPFNv2. Rather than proposing a new model architecture, the authors aim to dissect the mechanisms behind TabPFNv2's success and address its known limitations (e.g., performance on large-scale, high-dimensional, and multi-class datasets). The paper makes three main contributions: (1) it uncovers how TabPFNv2 internalizes attribute token learning despite randomized inputs; (2) it repurposes TabPFNv2 as a feature encoder via a novel leave-one-fold-out embedding extraction strategy; and (3) it introduces effective post-hoc, test-time divide-and-conquer extensions to improve scalability.

**Questions:**

- The introduction states 'TabPFN v2 internalizes attribute token learning to handle data heterogeneity.’: I could not understand what the new insight from the paper is. That TabPFN does prediction in a single forward pass was already a major selling point for the first TabPFN publication and the paragraph reads as if this is just reformulated. The explanations and illustrations in Section 5 are interesting, but don't seem like they present particularly new insights. That TabPFN(v2) can reliably infer meaningful attribute relationships through in-context learning was already presented in the original TabPFN paper. Could you briefly summarize the core insights that are novel?
- Line 60: 'Our findings reveal that TabPFN v2 effectively maps tabular instances into a nearly linearly separable embedding space' - Yes, TabPFNv2 maps instances into a linearly separable space. So does every other deep learning model since the last layer is always a linear layer with target-specific activation. Therefore, it is obvious that the embeddings are as close to being linearly separable as possible and that training a linear model on top yields strong performance. I did not understand what the new insights are, besides that out-of-fold embeddings need to be used?
- The random forest method proposed in 7.1 seems to be the same as the one used in the TabPFNv2 paper for handling large sample sizes, but in this case for features? Also the method introduced in 7.2 seems to be entirely taken from the TabDPT paper? Also the method to handle large datasets seems to be a variation of an existing method. It seems like no conceptually new proposal has been made?

**Ethical Concerns:**

["NO or VERY MINOR ethics concerns only"]

**Final Justification:**

The rebuttal has resolved some of my concerns, therefore, I update my score.

**Limitations:**

yes

**Paper Formatting Concerns:**

None.

**Quality:**

3

**Strengths And Weaknesses:**

- Understanding capabilities of prior fitted networks is an important and timely research topic, given their recent popularity
- The paper is well organized and clearly written.
- Extensive evaluation with several baselines and clearly described evaluation protocol.
- Careful considerations on dataset selection excluding datasets that were used for checkpoint selection by the TabPFN authors.
- The authors repeatedly emphasize that analytical studies are ‘more impactful than proposing yet another architecture.’ – I agree with that.
- The proposed extensions are practical, easy to apply, and follow successful practices in related applications (i.e. the use of out-of-fold data is used in state-of-the-art ensembling approaches for tabular data).

### Weaknesses
- The major weakness of the paper is that many of the insights seem obvious and are limited in novelty. Please refer to the questions section for details.
- A single validation and test set is not enough to establish statistical significance on dataset level, hence, conclusions based on dataset level results are not reliable.

---

> ### Author Rebuttal · Authors · 2025-07-31
>
> We sincerely thank Reviewer pvYD for the thoughtful review and for recognizing the importance and timeliness of our work, the clarity of our paper, and the rigor of our evaluation methodology.
>
> We respond to the specific questions and comments below:
>
> **Q1: Statistical significance and use of a single validation/test split**
>
> Thanks for the comment. We followed standard evaluation protocols from prior work (e.g., [Gorishniy et al., 2021]) to ensure comparability, using a 64/16/20 training/validation/test split. We agree that multiple train/validation/test splits would provide stronger statistical guarantees; however, this would incur a prohibitive computational cost due to the need for full hyperparameter tuning across all baselines and 273 datasets.
>
> To assess robustness, we performed an additional comparison using **5-fold cross-validation** on 10 classification datasets from the *Talent-tiny benchmark*. We observed consistent relative rankings across both hold-out and cross-validation settings:
>
> | Method     | Avg. Rank (hold-out) | Avg. Rank (5-fold CV) |
> |------------|-----------------------|------------------------|
> | TabPFN v2  | **1.86**                  | **1.64**                   |
> | ModernNCA  | 1.95                  | 2.64                   |
> | TabM       | 3.05                  | 2.45                   |
> | FTT        | 3.14                  | 3.27                   |
>
> These results confirm that the superiority of TabPFN v2 reported in our paper is robust to the evaluation protocol. We will include this consistency check in the final version.
>
> **Q2: Novelty of insights on randomized tokens and attribute token learning**
>
> We appreciate your comment. We want to first reiterate that our novel insight lies in understanding how “randomized tokens” work and enable TabPFN v2 to handle data heterogeneity (Lines 48 - 50). We certainly are aware that both TabPFN and TabPFN v2 adopt in-context learning and a single forward pass for inference (Line 89). However, their mechanism to handle data heterogeneity are drastically different (section 3).
>
> The original TabPFN v1 relied on padding features to a fixed dimension (e.g., 100) to preserve their original semantics, whereas **TabPFN v2 introduces randomized tokens**, sampled at inference time, for each attribute. This design eliminates the need to define attribute- or dataset-specific tokens, syntactically enabling direct application of the pre-trained model across heterogeneous datasets (section 5.1).
>
> Our novel insight is that, although randomized tokenization appears to discard attribute semantics, TabPFN v2 can *still* consistently infer meaningful attribute relationships through in-context learning. Specifically, we show that randomized tokens act like high-dimensional “continuous one-hot encodings” that reliably differentiate attributes across instances (Line 245). Pre-training on diverse tasks enables TabPFN v2 to infer co-occurrence patterns and relative attribute importance purely from the statistical structure of each dataset, without pre-defined attribute semantics.
>
> **Q3: Linearly separable embeddings and leave-one-fold-out strategy**
>
> We appreciate your comment. Before diving into the claim in line 60, we kindly invite the reviewer to revisit our preamble in lines 54 - 56. Just like what the reviewer argued, after seeing the superior performance of TabPFN v2, we expect to see a linear separable feature space produced by TabPFN v2, similar to other deep learning models. However, verifying, visualizing, and extracting it is non-trivial and not obvious (Lines 56 - 58).
>
> This is because **TabPFN v2 is a non-standard in-context learner**, and naive extraction of embeddings from its output tokens leads to surprisingly discrepant features between training and test instances—stemming from their different label embedding types (true vs. dummy). As demonstrated in Figure 4(b), embeddings extracted in this way create divergent clusters for training (darker crosses) and test (lighter circles) points, leading to poor generalization when used in a linear classifier.
>
> Our novelty lies in a carefully designed approach to extract features from TabPFN v2 (section 6.2). We propose a **leave-one-fold-out embedding extraction strategy** to resolve this issue: training instances are iteratively placed in the query set with dummy labels, while the remaining folds form the support set with true labels. This aligns the roles of training and test embeddings, resulting in a comparable embedding space. We empirically show that this strategy significantly improves linear probing performance (Table 2) and yields embeddings that match TabPFN v2’s in-context performance.
>
> In sum, our contribution lies not merely in observing separability but in **identifying and addressing a fundamental inconsistency in naive embedding extraction** that was not discussed in prior work.
>
> **Q4: Extensions may overlap with existing methods (e.g., Random Forest, TabDPT)**
>
> We appreciate your comment, and we emphasize the differences and our contributions as follows. First of all, the purpose of section 7 is to demonstrate that the limitations of TabPFN v2 (Lines 13 - 15 and 65 - 71) can be effectively addressed without re-training the model, reminiscent of test-time scaling. In our humble opinion, our key novelty and contribution lie in pointing out and demonstrating this concept, not the particular methods we applied. Indeed, we deliberately choose not to propose a completely new mechanism but borrow and adapt existing mechanisms originally proposed in different contexts to TabPFN v2.
>
> - **Large-scale datasets**:
>
>  The decision tree integration in TabPFN v2’s original paper is used for *small-and medium-scale datasets suffering from within-dataset heterogeneity*. We adapt this approach for a **different purpose**—handling large-scale datasets by partitioning them into smaller subsets where TabPFN v2 can operate effectively.
>
> - **Many-class classification**:
>
> We have provided the detailed implementation and the difference between ours and TabDPT (denoted as PFN-DPT in our paper) in Appendix C lines 92-99.
> While decimal encoding (TabDPT) is an existing idea, our TabPFN v2∗ improves robustness by *randomly permuting class-to-digit mappings √C times* and ensembling predictions. This mitigates artificial correlations introduced by fixed decimal encodings. We ensure fair comparison by matching the number of inferences of TabPFN v2 when ensembling their results in different methods.
>
> | Method               | Mean Accuracy |
> |----------------------|----------------|
> | TabDPT (no ensemble) | 0.6047         |
> | TabDPT (ensemble)    | 0.6567         |
> | TabPFN v2∗ (ours)    | **0.7093**         |
> Our method is superior, and we explicitly state in the paper that TabPFN v2-DPT (Figure 5, middle) refers to the ensemble variant.
>
>
> - **High-dimensional datasets**:
>
> We adapt Random Forest-style feature subsampling and bagging specifically for TabPFN v2 (Section 7.1), a direction unexplored in the original paper.
>
> In light of our further clarification, we would appreciate it if the reviewer would reconsider the rating. Thank you.

---

> > ### Comment · Reviewer_pvYD · 2025-08-06
> >
> > Thank you for your detailed rebuttal and for taking the time to respond to my concerns.
> >
> > The novelty concerns in Q2 have been resolved. Regarding Q3: using leave-one-fold-out is the standard approach for generating features with target involvement. This is done i.e. for leave-one-out-encoding of categorical features or with the predictions of many models in multi-layer stacked ensembling. Therefore, I still struggle to see the methodological novelty. However, I agree that the practical demonstration is useful.
> >
> > I also agree that regardless of the methodological novelty, demonstrating the concept of improvements without retraining is valuable. However, I want to point out that this is not new knowledge, as it has been already done in the TabPFNv2 paper. Given that the experiments are still insightful and the rebuttal resolved some of my concerns, I will revise my score.

---

> > > ### Author Response · Authors · 2025-08-07
> > > **Re: Official Comment by Reviewer pvYD**
> > >
> > > We sincerely thank the reviewer for the thoughtful follow-up and for the willingness to revise the score based on our clarifications.
> > >
> > > We agree that the leave-one-fold-out (LOFO) strategy has been adopted in classical machine learning settings. Our intent was not to present LOFO itself as a methodological novelty, but rather to demonstrate its effectiveness in a new application context — extracting features for TabPFN v2, whose in-context inference (with dummy label tokens assigned to test instances) inherently makes training and test embeddings misaligned. We are glad that the reviewer found the demonstration practical and useful, and we will further clarify this in the final version.
> > >
> > > Regarding improvements without retraining (Section 7), we acknowledge that TabPFNv2 has performed a preliminary exploration in this aspect (as mentioned in our response to Q4). We are grateful that the reviewer recognized the practical value of our further exploration and extension that considers multiple scenarios (large-scale, high-dimensional, and many-class). We will surely incorporate our rebuttal into the final revision for clarity.

---

> ### Author Response · Authors · 2025-08-06
>
> Dear Reviewer,
>
> Thank you for taking the time to review our submission. We sincerely appreciate your feedback and have done our best to address your comments and concerns in the author response.As the discussion period is ongoing, we would like to kindly check in to see if you had a chance to review our rebuttal and whether you have any further questions or require clarifications. We are happy to provide additional information or engage in discussion to help improve the quality and clarity of the paper.
>
> Thank you again for your time and effort in the review process.
>
> Best regards,
>
> Authors

---

### Official Review · Reviewer_wWLo · 2025-07-03

**Clarity:** 3
**Significance:** 2
**Originality:** 3
**Rating:** 5
**Confidence:** 5

**Summary:**

The paper provides an in-depth analysis on the recent tabular foundation model, TabPFNv2. The results show characteristics of TabPFNv2 that can aid strong performances, and be used as a feature extractor. Moreover, the paper addresses some limitations and approaches to overcome them.

**Questions:**

Please refer to weaknesses.

**Ethical Concerns:**

["NO or VERY MINOR ethics concerns only"]

**Final Justification:**

The authors have addressed many of my concerns and added empirical studies that could strengthen the quality of the paper.

**Limitations:**

It could be interesting to elaborate more on some possible extensions for repurposing the extracted features through feature encoding of TabPFNv2, and some more analysis with regression tasks.

**Paper Formatting Concerns:**

No concerns for paper formatting.

**Quality:**

3

**Strengths And Weaknesses:**

Strengths:
- The paper is generally well-written and easy to follow.
- The paper provides insights on important factors for in-context learning tabular foundation models.
- The paper provides some solid directions for overcoming the limitations of TabPFNv2.

Weakness:
- In regards to the point 'internalizes attribute token learning', it could be interesting to compare it with the TabICL, which might be doing something similar in a different way, and emphasize the importance of this mechanism for in-context tabular foundation models.
- On the results on section 6.3, it would be interesting to see results on regression problems with simple linear models.
- Also, in regards to the feature encoder, it could be useful to note that the work may be useful to integrate with other modules, such as domain adaptations methods.
- Overall, how would or what are the (expected) results if TabPFNv2 is fine-tuned?
- It could be interesting to compare with TabICL [1] on 'Multi-Class Problems with More Than 10 Classes' (Or mention the work, if it has similar strategies)
- On a similar topic, it would be interesting to compare with TabICL for larger than 10K samples.

[1] Qu, Jingang, et al. "Tabicl: A tabular foundation model for in-context learning on large data." arXiv preprint arXiv:2502.05564 (2025).

---

> ### Author Rebuttal · Authors · 2025-07-31
>
> We sincerely thank Reviewer wWLo for the thoughtful and constructive feedback. We address each point below:
>
> **Q1: Comparison with TabICL and internalized attribute token learning**
>
> Thank you for the suggestion. We were aware of TabICL and have cited it [35], and we will extend the discussion. Both TabPFN and TabICL leverage in-context learning for tabular prediction, but they adopt distinct approaches to attribute tokenization. (Even TabPFN v1 and v2 adopt quite different tokenization mechanisms; see Lines 90 - 93 & sect. 3.) TabPFN v2 uses randomized attribute tokens (RTs), re-sampled per inference, which internalize feature relationships without requiring semantic alignment (Lines 207 - 214). In contrast, TabICL employs set transformers to encode distributional properties of an attribute, combined with RoPE for positional differentiation. While TabICL requires specifically learning distribution-aware tokens, TabPFN v2’s RTs enable plug-and-play generalization across heterogeneous tasks. We will expand this comparison in the final version to highlight the conceptual and practical implications of each approach.
>
> **Q2: Regression results and extensions for feature reuse**
>
> We have extended our feature extraction strategy to regression tasks. The embeddings generated via our leave-one-fold-out method show simplified and structured distributions, similar to the classification case. (Figures will be included in the final version.) To assess their utility, we trained linear regression (LR) on raw features, vanilla embeddings (c.f. section 6), and our leave-one-fold-out extracted embeddings. The average rank across regression datasets is summarized below:
>
> | Embedding           | Regressor | Avg. Rank |
> |---------------------|-----------|-----------|
> | Raw                 | LR        | 2.42      |
> | Vanilla    | LR        | 2.97      |
> | Ours                | LR        | 1.58      |
>
> Our embeddings lead to **better regression generalization than raw features and vanilla TabPFN features**. These results confirm that the extracted embeddings can serve as high-quality, reusable features.
>
> **Q3: Integration with domain adaptation methods**
>
> We agree this is a promising direction. Our feature extraction method offers a way to generate robust embeddings without retraining, which could complement domain adaptation techniques and enable cross-domain generalization. We will highlight this in future discussions.
>
> **Q4: Fine-tuning TabPFNv2**
>
> Thank you for the question. We note that a recent concurrent study ([Rubachev et al., 2025]) explores fine-tuning TabPFN v2 and demonstrates improvements using simple full finetuning. In our analysis (Table 2), we observe that combining embeddings from multiple layers enables strong linear probing performance—outperforming the original TabPFN output. This suggests that probing can serve as a lightweight, interpretable alternative to fine-tuning.
>
> However, fine-tuning TabPFN v2 remains challenging for high-dimensional or many-class tasks due to (a) the scarcity of high-dimensional datasets and (b) the architectural constraint that TabPFN v2 is designed as a 10-class output format. To adapt it to many-class problems, one must jointly fine-tune both the label encoding scheme and the output head—adding substantial complexity and reducing generality. In contrast, our post-hoc divide-and-conquer strategies handle these challenges effectively without retraining, aligning with recent trends in test-time scaling in LLMs.
>
> For large-scale datasets, fine-tuning is an effective choice given enough training time. Due to rebuttal time constraints, we fine-tuned TabPFN v2 on two large-scale datasets and compared it with our post-hoc divide-and-conquer strategies.
>
> | Dataset            | Method        | Accuracy (%) | Training Time (s) |
> |--------------------|---------------|---------------|---------------------|
> | **nomao** (34K × 118) | PFN v2        | 96.08         | 9.2                 |
> |                    | PFN v2*-DF (ous)  | 96.75         | 8.0                 |
> |                    | Fine-tuned    | 97.01     | 1771                |
> |                    | CatBoost      | 97.03         | –                   |
> | **jannis** (83K × 54) | PFN v2        | 70.24         | 37.9                |
> |                    | PFN v2*-DF (ous)   | 72.57         | 17.7                |
> |                    | Fine-tuned    | 73.77    | 12201               |
> |                    | CatBoost      | 72.26         | –                   |
>
> These results show that while fine-tuning can yield marginal performance gains, it comes with a significant computational cost. In contrast, our PFN v2*-DF approach achieves competitive performance **with substantially lower overhead** by taking advantage of its "divide-and-conquer" strategy, offering a practical alternative for scaling TabPFN v2.
>
> **Q5: Comparison with TabICL on many-class classification**
>
> We followed the setup in Figure 5 (middle) to evaluate TabICL on multi-class datasets (>10 classes). The average accuracy of TabICL is 0.6901, while our method achieves 0.7093. These results suggest our strategy scales more effectively to complex label spaces. We will extend this discussion in the final version.
>
> **Q6: Comparison with TabICL on large-scale datasets (>10K samples)**
>
> Using the configuration in Figure 5 (right), we report the average ranks on the large-scale classification datasets:
> - PFNv2*-DF: 4.50 (with 4 ensemble models)
> - TabICL: 6.25 (with 4 ensemble models)
>
> Our method demonstrates stronger generalization in large-scale settings, highlighting its scalability advantages. We will extend this discussion in the final version.
>
> In light of our further clarification, we would appreciate it if the reviewer would consider increasing the rating. We would be happy to address additional questions you may have in the author-reviewer discussion phase.

---

> > ### Comment · Reviewer_wWLo · 2025-08-07
> > **Thank you for the sincere rebuttal.**
> >
> > Thank you for the sincere rebuttal. It has resolved many of the concerns that I had in mind.
> >
> > For the linear regression part, if time is allowed, it could be useful to try the efficient Ridge (possibly RidgeCV in sklearn) to add some regularization.

---

> ### Author Response · Authors · 2025-08-06
>
> Dear Reviewer,
>
> Thank you for taking the time to review our submission. We sincerely appreciate your feedback and have done our best to address your comments and concerns in the author response.As the discussion period is ongoing, we would like to kindly check in to see if you had a chance to review our rebuttal and whether you have any further questions or require clarifications. We are happy to provide additional information or engage in discussion to help improve the quality and clarity of the paper.
>
> Thank you again for your time and effort in the review process.
>
> Best regards,
>
> Authors

---

> ### Author Response · Authors · 2025-08-07
> **Thanks for the feedback and suggestion from Reviewer wWLo**
>
> Thank you for the thoughtful suggestion and for your encouraging feedback. Following your advice, we replaced the original Linear Regression with RidgeCV from scikit-learn for regression tasks based on the extracted features. We are pleased to report that this regularization indeed yields notable improvements (for all feature extraction methods). More importantly, using our feature extraction method still yields the lowest error, suggesting its consistent effectiveness. Below are the updated average ranks across benchmark datasets:
>
> | Embedding | Regressor | Avg. Rank |
> |-----------|-----------|-----------|
> | Vanilla   | LR        | 5.58      |
> | Vanilla   | RidgeCV   | 5.08      |
> | Raw       | LR        | 3.83      |
> | Raw       | RidgeCV   | 3.17      |
> | Ours      | LR        | 2.08      |
> | Ours      | RidgeCV   | 1.25      |
>
> We will incorporate these updated results into the final version of the paper. Once again, thank you for your valuable input—it has helped strengthen the overall quality of our work.

---

### Decision · Program_Chairs · 2025-09-17

**Decision:**

Accept (poster)

**Comment:**

This paper provides an empirical and analytical examination of TabPFNv2. The work focuses on analyzing the factors underlying TabPFNv2’s performance and addressing limitations related to large-scale, high-dimensional, and multi-class datasets. The contributions are threefold: (1) demonstrating how TabPFNv2 learns attribute tokens even with randomized inputs; (2) adapting TabPFNv2 as a feature encoder through a leave-one-fold-out embedding; and (3) proposing test-time divide-and-conquer extensions to improve scalability.

Afer the discussion, most of reviewers' concerns are addressed and all recommended for acceptance.